



# Species richness and functional attributes of fish assemblages across a large-scale salinity gradient in shallow coastal areas

Birgit Koehler[1], Mårten Erlandsson[1], Martin Karlsson[2], Lena Bergström[1]

[1]Department of Aquatic Resources, Institute of Coastal Research, Swedish University of Agricultural Sciences, Skolgatan 6, 74243 Öregrund, Sweden

[2]Department for Marine Management, The Swedish Agency for Marine and Water Management, Gullbergs Strandgata 15, 41104 Gothenburg, Sweden

Correspondence to: Birgit Koehler (birgit.koehler@slu.se) or Lena Bergström (lena.bergstrom@slu.se)

**Abstract.** Coastal ecosystems are biologically productive and their diversity underlies various ecosystem services to humans. However, large-scale species richness (SR) and its regulating factors remain uncertain for many organism groups, owing not least to the fact that observed SR ($SR_{obs}$) is strongly dependent on sample size and inventory completeness (IC). We estimated changes in SR across a natural geographical gradient using statistical rarefaction and extrapolation methods, based on a large fish species incidence dataset compiled from Swedish fish survey databases. The data covered nearly five decades (1975-2020), a 1,300 km north-south distance and a 10-fold salinity gradient along sub-basins of the Baltic Sea plus Skagerrak. Focusing on shallow coastal and offshore areas (<30 m depth), we calculated standardized SR ($SR_{std}$) and estimated SR ($SR_{est}$), and related these to sub-basin annual mean salinity and water temperature. IC was high, 98.5% - 99.9%, in the 10 sub-basins with sufficient data for analysis. The recorded fish species were of 75% marine and 25% freshwater origin. Total fish $SR_{obs}$ was 144 for shallow coastal areas, and 110 for shallow offshore areas. Sub-basin specific $SR_{est}$ for coastal areas varied between $35 \pm 7$ (SE) and $109 \pm 6$ fish species, and was ca. three times higher in the most saline (salinity 29-32) compared to the least saline sub-basins (salinity 2.7). Completing information on functional attributes showed that differences along the salinity gradient reflected an increased share of coastal resident fish species in lower salinities, and a higher share of migratory fish at higher salinities. The proportion of benthic and demersal fish species was also lower in the least saline sub-basins, and increased with increasing salinity. If climate change lowers the salinity regime of the Baltic Sea in the future this may hence influence the SR and community composition of fish.



# 1 Introduction

Biodiversity is essential for ecosystem processes, and ultimately for the humans depending on these (IPBES, 2019). Coastal ecosystems are often biologically diverse and highly productive, providing valuable ecosystem services to humans, such as food, water purification and protection against floods (Griffiths et al., 2017; Kraufvelin et al., 2018; Pan et al., 2013). However,

threats to coastal biodiversity from e.g. overfishing, habitat loss, pollution, eutrophication and climate change are many and profound (Duncan et al., 2015; Griffiths et al., 2017; Pan et al., 2013). At the same time, actually occurring coastal species numbers often remain uncertain (Appeltans et al., 2012). This makes improved understanding of their biodiversity especially important to support conservation and management measures (Pan et al., 2013; Rooney & McCann, 2012).

Taxonomic inventories, or species censuses, are required e.g. for the analysis of biodiversity patterns, delineation of species

ranges, and prioritization of conservation efforts (Mora et al., 2008). Species richness (SR), i.e. the number of species in an ecosystem, is a classical indicator of biodiversity, also referred to as "alpha diversity" (Gotelli & Colwell, 2001; Hill, 1973). However, since achieving complete species inventories is often impracticable with realistic sample efforts, most censuses remain incomplete and many rare species remain unknown. Consequently, it is important to consider the effect of sample size and inventory completeness (IC) on observed SR ($SR_{obs}$) to avoid biased or misleading comparisons or interpretations (Chao

& Chiu, 2016; Chao et al., 2020; Colwell & Coddington, 1994; Mora et al., 2008).

SR is connected to several ecosystem processes, such as productivity (Duffy et al., 2017), and the efficiency of resource use and nutrient cycling. SR may also facilitate the simultaneous provision of several ecosystem processes, i.e. an ecosystem's multifunctionality (Byrnes et al., 2014). However, since species do not contribute equally to ecosystem functioning, the diversity of species functional attributes adds another important dimension to ecosystem understanding (Duncan et al., 2015;

Reiss et al., 2009). Functional diversity can enhance long-term stability, through functional redundancy and complementarity, and can help to buffer ecosystems against disturbances (O'Gorman et al., 2011).

Salinity is a key variable influencing SR in coastal areas, as natural differences in salinity among locations function as a threshold or "ecological barrier" for the distribution of freshwater and marine species (Olenin & Leppäkoski, 1999; Vuorinen et al., 2015). At the same time, an intensified water cycle caused by global warming is currently changing the salinity regimes

of marine and coastal ecosystems (Durack et al., 2012; Liblik & Lips, 2019; Meier et al., 2021). It is important to understand how salinity influences species' distributions in aquatic ecosystems to be able to better predict how potential changes may affect ecosystem functioning.

The Baltic Sea, one of the world's largest brackish water bodies, exhibits a pronounced, geographically stable salinity gradient that is maintained by sporadic inflows of saline water from the North Sea through the Danish Straits and by freshwater input

from large rivers, especially in the north. Hence, the Baltic Sea gradient can serve as model on the influence of salinity on



species distributions (Johannesson & Andre, 2006; Ojaveer et al., 2010), that has been studied for various organism groups. $SR_{obs}$ was often higher at the more saline conditions, e.g. for benthic bacteria, benthic macroalgae and benthic meio- and macrofauna (Broman et al., 2019; Klier et al., 2018; Middelboe et al., 1997). In other studies, $SR_{obs}$ was highest at highest salinity, lowest at intermediate salinity and intermediate at lowest salinity, e.g. for phytoplankton and benthic macrofauna
(Bonsdorff, 2006; Olli et al., 2019; Zettler et al., 2014), or there was no clear trend between $SR_{obs}$ and salinity, e.g. for bacterio-, pico- and mesoplankton (Herlemann et al., 2016; Hu et al., 2016).

The species composition of fish in the Baltic Sea is regulated by salinity as well (Olsson et al., 2012; Pekcan-Hekim et al., 2016), with fish $SR_{obs}$ generally being higher at higher compared to lower salinities (HELCOM, 2020; Hiddink & Coleby, 2012; Lappalainen et al., 2000; MacKenzie et al., 2007; Ojaveer et al., 2010; Pecuchet et al., 2016; Thorman, 1986). Various
studies have also reported changes in fish $SR_{obs}$ or species composition over time (e.g. Ammar et al., 2021; Törnroos et al., 2019). However, despite concerns that fish SR may decline in the future due to decreasing upper layer salinity (e.g. MacKenzie et al., 2007; Pecuchet et al., 2016; Vuorinen et al., 2015), information on how the complete coastal fish assemblage varies spatially in relation to the Baltic Sea salinity gradient, including potential differences across functional groups, is lacking. Hence, there is a need to complement already existing information on the influence of salinity on various Baltic Sea organism
groups with more complete information in relation to fish diversity, taxonomically and functionally. This kind of understanding for multiple trophic levels is needed to better understand and predict how changing salinity, in the Baltic Sea and in coastal areas in general (Durack et al., 2012; Liblik & Lips, 2019), may affect ecosystem structure and functioning (MacKenzie et al., 2007). For example, if different species groups are differently affected, this may also change biotic interactions such as benthic-pelagic coupling, with effects on exchanges of energy, mass or nutrients between benthic and pelagic habitats (Griffiths et al.,
2017). Moreover, understanding species richness at a broader, sub-regional scale is important to support analyses of potential species richness and species compositions at more local scales within each sub-basin.

To this aim, we compiled a large dataset on fish species observations in shallow (<30 m depth) Swedish coastal and offshore areas, based on multiple existing sources of Swedish mapping and monitoring combined over the years 1975-2020. The extensive dataset covered fish species incidence information from 1,848 unique observations/fishing occasions, during which
in total 24,415 species incidences were recorded. Geographically, the data covered 12 hydrographically distinct sub-basins, and a ca. ten-fold salinity gradient from close to freshwater conditions in the inner Baltic Sea to close to fully marine conditions at the Swedish west coast. Since $SR_{obs}$ is strongly dependent on sample size, which differed between sub-basins, we used statistical rarefaction-extrapolation methods to estimate IC and standardized SR per sub-basin. Further, we categorized each fish species according to origin (marine vs. freshwater) as well as three functional attributes based on coastal habitat preference,
vertical preference and feeding habitat, and investigated the influence of salinity (and, for comparison, temperature) on fish SR in total and within the functional attributes. We discuss the results in the context of the regulating influence of salinity on





fish SR and community composition in coastal ecosystems, and potential implications for conservation and ecosystem management.

## 2 Methods

### 2.1 Study system

The Baltic Sea, an enclosed, essentially non-tidal brackish marine region with a maximum and mean depth of 460 and 54 m, respectively, and a water residence time of 25-40 years, is, among the world's largest estuaries (area: 415,000 km$^2$; HELCOM, 2018). Its current brackish conditions were formed by gradual narrowing of its opening to the North Sea and have been in place since ca. 3,000 years (Russell, 1985). Due to its geographically variable but locally relatively stable salinity conditions

the Baltic Sea has been called a "marine-brackish-limnic continuum" (Bonsdorff, 2006). Its surface salinity changes from <3 (psu) in the inner-most areas in the north and north-east to almost fully marine (ca. 29) in the Kattegat in the southwest (Table 1). Within this gradient, the Baltic Sea can be divided into hydrographically distinct sub-basins, mostly separated by shallow sounds or sills. To strengthen the database with respect to higher salinity areas we additionally included a North Sea sub-basin adjacent to Kattegat, i.e. Skagerrak (salinity ca. 30; Table 1).

**Table 1. Salinity and temperature in Swedish coastal areas, given as mean (± SE) annual values per sub-basin across the years 1993-2019. Values represent conditions by the bottom at 0-30 m depth based on data from the EU Copernicus Marine Service Information (CMEMS, 2021).**

| Sub-basin | Salinity | Temperature (°C) |
|---|---|---|
| Bothnian Bay | 2.68 ± 0.01 | 4.53 ± 0.23 |
| The Quark | 4.26 ± 0.01 | 5.38 ± 0.25 |
| Bothnian Sea | 5.10 ± 0.01 | 5.44 ± 0.22 |
| Åland Sea | 5.80 ± 0.01 | 6.44 ± 0.25 |
| N Baltic Proper | 6.37 ± 0.01 | 6.43 ± 0.22 |
| E Gotland Basin | 6.85 ± 0.01 | 7.30 ± 0.24 |
| W Gotland Basin | 6.88 ± 0.01 | 6.48 ± 0.20 |
| Bornholm Basin | 7.60 ± 0.02 | 8.15 ± 0.24 |
| Arkona Basin | 10.96 ± 0.07 | 8.92 ± 0.26 |
| The Sound | 23.42 ± 0.14 | 9.72 ± 0.24 |
| Kattegat | 29.02 ± 0.05 | 9.32 ± 0.21 |
| Skagerrak | 32.40 ± 0.03 | 9.62 ± 0.22 |

Reflecting its salinity conditions the Baltic Sea harbors a unique fish fauna with a mixture of freshwater species (e.g. pike, perch, pikeperch), and marine species (e.g. cod, herring; (Olsson et al., 2012). Further, many marine fish populations have



adapted to the brackish conditions from their Atlantic counterparts (Laikre et al., 2005), for example Baltic cod and herring populations, and one flounder species is endemic to the Baltic Sea (Momigliano et al., 2018). Hence, the Baltic Sea may also have a unique value as a refuge for evolutionary lineages, and constitute an important genetic resource for management and conservation (Johannesson & Andre, 2006).

## 2.2 Species richness data

The primary source of fish species data was the Swedish National database of coastal fish ([www.slu.se/kul](www.slu.se/kul)), which holds data from surveys encompassing coastal fish monitoring, mapping projects and surveillance programs over the entire salinity gradient of the Baltic Sea plus Skagerrak. Coastal areas were delineated using official national definition. Data from shallow depths < 30 m were selected, corresponding to the main represented sampling methods in the database (Table S1), and with some margin to the photic depth in the concerned coastal habitat types (Kaskela et al., 2012). Only sampling occasions with

geographical coordinates and verified sampling references were included, giving 154,172 data entries, i.e. individual fish that had been caught and determined to species. The size of the coastal shallow areas ranged from 240 km$^2$ (Åland Sea) to 5,798 km$^2$ (Bothnian Bay; Table 2). Corresponding data from shallow offshore areas (< 30 m) were also compiled for comparison (5,601 data entries). Further, additional data from 1) a national coastal trawl survey ($n$=4,420 for coastal and $n$=382 for offshore areas), 2) the ICES-coordinated International Bottom Trawl Survey ($n$=1,969 for coastal and $n$=2,099 for offshore areas) and

3) national projects using standardized methodology ($n$=893 for coastal areas), all carried out in the Skagerrak, Kattegat and The Sound, were included, selecting only hauls from <30 m depth within the concerned geographical delineations. Corresponding trawl data (<30 m) are not collected in Swedish waters of the other sub-basins.

Hence, data collected from multiple gears were combined, including gill nets, fyke nets, seines, trap nets, low impact underwater detonations and trawls, in order to maximize the chance of including different species (Table S1). The ambition to

collate information from all available fish surveys implied some differences in predominating data collection methods across the studied geographical range. The main data sources were trawls and trap net surveys in the most saline sub-basins, i.e. Skagerrak, Kattegat and The Sound, and gill net surveys in the remaining sub-basins (Table S1). The analytical approach was chosen to encompass this variability when making comparisons among sub-basins (see Sects. 2.3 and 4).

Observed SR, SR$_{obs,}$ was reported for all sub-basins, but statistical analyses and comparisons were conducted only for sub-

basins containing data from at least 25 sampling/fishing occasions. This was the case for ten coastal sub-basins (Bothnian Bay, The Quark, Bothnian Sea, Åland Sea, N Baltic Proper, W Gotland Basin, Bornholm Basin, The Sound, Kattegat and Skagerrak), and one off-shore sub-basin (Kattegat). This dataset is referred to as "raw data", and contained 160,453 entries (i.e. fish individuals caught and determined to species) from 1,638 sampling/fishing occasions at 4,571 unique locations for shallow coastal areas, and 2,762 entries from 137 sampling/fishing occasions at 199 unique locations for shallow offshore

areas.



Moreover, we searched for evidence of fish species that had remained undetected in our fish incidence database, by identifying fish species records from three additional sources, using the same criteria for geographical and depth delineations as above, i.e. 1) the SLU hosted national public database for citizens' reporting of species observations (SLU Swedish Species Information Centre, https://www.artportalen.se/; $n$=8,926 for coastal and $n$=290 for offshore areas after unreasonable species

observations were discarded), 2) the national archive for oceanographic data hosted by the Swedish Meteorological and Hydrological Institute (SHARKweb, https://www.smhi.se/en/services/open-data/national-archive-for-oceanographic-data; $n$=1,259 for coastal and $n$=135 for offshore areas), and 3) published inventory data for Swedish shallow coastal areas in Skagerrak, Kattegat and Bornholm Basin (Pihl & Wennhage, 2002; Pihl et al., 1994; Wikström A., 2009). This "additional data sources" were used as complementary information on SR$_{obs}$ but could not be used in the statistical analysis since they

only included presence-information for the reported species, rather than complete sampling and species incidence information. Further, our SR results were compared with the HELCOM (2020) checklist on macro-species containing information for all of the sub-basins and depths in the Baltic Sea region.

## 2.3 Analysis of species richness data

The raw data was first summarized to a dataset of *unique fish species* caught per fishing/sampling occasion in presence/absence

format, and then further aggregated to an incidence frequency format, giving the observed total incidence of each species over the number of fishing/sampling occasions. This dataset is referred to as "fish incidence database". Each unique combination of a fishing/sampling location per date was defined as one sampling unit, and these were summed per sub-basin to obtain the sample sizes. Subsequently, incidence-based Hill diversity numbers of three orders, which differ in their propensity to include or exclude relatively rarer species (Hill, 1973), were calculated to quantify the species diversity of each assemblage, i.e. 1)

species richness (SR), which counts all species equally irrespective theirincidence frequency, 2) Shannon diversity (ShD), which considers the incidence frequency and can be interpreted as the effective number of frequent species, and 3) Simpson diversity (SiD), which can be interpreted as the effective number of highly frequent species (Chao et al., 2014; Chao et al., 2020; Hill, 1973). Calculations were performed using the R package *iNEXT* and the functions *ChaoRichness*, *ChaoShannon* and *ChaoSimpson* (Chao et al., 2020; Hsieh et al., 2016), and the Hill numbers are hereafter referred to as *observed* SR, ShD

and SiD, respectively. It should be noted that, using these methods, Shannon and Simpson diversity are expressed in terms of richness, i.e. number of species, which differs from other known formats.

SR$_{obs}$ is highly dependent on sample completeness (Colwell & Coddington, 1994; Hill, 1973) and may typically underestimate the true SR due to undetected species (also referred to as under-sampling, sampling bias or sampling problem; Chao et al., 2014; Chao & Jost, 2015; Menegotto & Rangel, 2018). We used coverage-based rarefaction and extrapolation methods to

correct for this effect (Chao & Jost, 2012). The Chao richness method, a non-parametric asymptotic richness estimator that is based on the frequency of rare species in the sample (Chao et al., 2014), was used to estimate the actual, asymptotic fish SR for each sub-basin (*ChaoRichness* function in the R package *iNEXT*; Hsieh et al., 2016), and the estimated parameters were



interpreted as described and exemplified in (Chao et al., 2020). The respective values are hereafter referred to as *estimated* values (i.e. $SR_{est}$, $ShD_{est}$ and $SiD_{est}$). Inventory (sample) completeness (IC) was calculated based on sample coverage (Chao &
Jost, 2012; Hsieh et al., 2016). To compare data across sub-basins, SR, ShD and SiD were standardized to the minimum observed IC across sub-basins (*estimateD* function in the R package *iNEXT;* Hsieh et al., 2016). The respective values are hereafter referred to as *standardized* values (i.e. $SR_{std}$, $ShD_{std}$ and $SiD_{std}$). Similar analyses were also conducted for SR of fish with different functional attributes (see Sect. 2.4). All calculations were conducted using R version 4.0.4 (R Core Team, 2021).

### 2.4 Fish functional attributes

All observed fish species were assigned functional attributes based on ecological and behavioral traits, as well as into being of either marine or freshwater origin (Kullander, 2002). The affinity of each species to different parts of the coastal habitat, or habitat preference, was assigned based on (Elliott & Dewailly, 1995; Pihl & Wennhage, 2002), however with certain adaptations to suit both marine and brackish conditions (Table S3-S6). Applied categories were: Catadromous or anadromous migrants (CA), using coastal habitats only when migrating between marine and freshwaters for spawning and feeding; Marine
juvenile migrants (MJ), using coastal habitats primarily as nursery or feeding grounds; Marine visitors (MV), occurring irregularly in the coastal area, having their primary habitat in deeper waters; Marine seasonal migrants (MS), making regular seasonal visits to coastal habitats, usually as adults; and Coastal residents (CR), spending almost their complete life cycle in coastal habitats or the littoral coastal zone. The main vertical distribution of each species in the water column, considering the adult stage, was assigned based on (Elliott & Dewailly, 1995; Koli, 1990) as: Pelagic (P), living mainly in the water column;
Demersal (D), mainly associated with the bottom substrate; Demersal-pelagic (DP), alternating between the water column and bottom substrate; and Benthic (B), staying close to the seabed. Main feeding habits were assigned by combining information on feeding guild (Elliott & Dewailly, 1995) with trophic levels (TL) and principal diet composition (Froese and Pauly, 2021), as: Piscivores (Pi; TL 3.6 - 4.4); Invertebrate and fish eaters (IF; TL 2.9 - 3.9); Invertebrate eaters (I; TL 2.8 – 3.9); Planktivores (PL; TL 3.1 - 3.2) and Omnivores (O; TL 2.8 - 3.5).

**2.5 Sea water salinity and temperature**

For each sub-basin, data on ambient salinity and temperature was extracted from the "Baltic Sea Physics Reanalysis" product, as calculated by the Swedish Meteorological and Hydrological Institute (SMHI) with the coupled physical-biochemical model system NEMO-SCOBI, and available from year 1993 (CMEMS, 2021). This encompassed full coverage layers with a 4 km x 4 km grid. Monthly mean values close to the sea bed for all grid cells representing areas less than 30 m depth were first
identified, and then used for calculating long-term means and standard deviations for the years 1993-2019.



**2.6 Statistical analyses**

Linear regressions were used to analyze the relationships between salinity and temperature, respectively, and observed, standardized and estimated SR, ShD and SiD. To test for any additional explanatory effect of temperature, after accounting for the effect of salinity, we used ANOVA to compare models with salinity as the only explanatory factor with models with

salinity plus temperature as explanatory factors. Furthermore, relationships were tested between the different functional attributes and salinity. To reduce skewness and approximate normality, left-skewed response variables were $\log_{10}$-transformed prior to analysis, or, in two cases where the response variable included zero-values, Yeo-Johnson transformed (Yeo and Johnson, 2000). All analyses were conducted using R version 4.0.4 (R Core Team, 2021).

**3 Results**

**3.1 Salinity and temperature**

The annual mean salinity varied more than ten-fold in shallow coastal areas across the studied sub-basins, from 2.7 in the northernmost Baltic Sea to 32.4 in the Skagerrak. Across the same geographical range, the annual mean water temperature varied from 4.5°C in the north to ca. 9-10°C in the Sound and outwards (Table 1).

**3.2 Fish species observations and distribution**

$SR_{obs}$ varied from 23 (Arkona Basin) to 101 (Kattegat) in shallow coastal areas (Table 2, that also contains related information on e.g. sample size and species incidences per sub-basin), and amounted to 125 across sub-basins and years. Since IC was <100% (see Sect. 3.3), this can be assumed a lower bound estimate of the true SR. Indeed, the additional data sources contained 19 more species that were not represented in the fish incidence database, resulting in a total fish $SR_{obs}$ of 144 in coastal areas (Table S3). Of the species in the fish incidence database, 49% occurred only in the higher salinity Skagerrak-Kattegat region

including The Sound, 15% occurred only in the Baltic Sea region (i.e., inside The Sound), and 36% occurred in both these regions. The most widely ranging speciescoastal area were herring (*Clupea harengus*), brown trout (*Salmo trutta*), European sprat (*Sprattus sprattus*) and eelpout (*Zoarces viviparous*), with incidences reported from all 12 sub-basins (Table S2).

For shallow offshore areas, $SR_{obs}$ varied from 11 (N Baltic Proper) to 74 (Kattegat; Table 2), and amounted to 96 across sub-basins and years. The additional data sources contained information on 14 more species, resulting in a total fish $SR_{obs}$ of 110

(Table S3). Of the species in the fish incidence database, 48% occurred only in the higher salinity Skagerrak-Kattegat region including The Sound, 21% occurred only in the Baltic Sea region, and 31% occurred in both regions. Herring was the only species reported in all the nine sub-basins for which fish incidence data for shallow offshore areas was available (Table S4).



**Table 2. Summary information and statistics for the fish incidence database and additional data sources, representing the 12 sub-basins and separately for Swedish shallow coastal and offshore areas (<30 m depth: Observations and estimated inventory completeness (IC) are given for all sub-basins. Standardized (SR$_{std}$) and estimated (SR$_{est}$) values were calculated for sub-basins with sample size >25 fishing/sampling occasions. SR$_{std}$ was calculated for an IC of 98.5%, which was the lowest IC among sub-basins with sufficient data (i.e. Åland Sea coastal areas: For comparison, SR$_{obs}$ if also including presence information from additional data sources (not included in the statistical analyses, see Sect. 2.2), and for whole sub-basins in the Baltic Sea according to HELCOM (2020, across countries and depths) are given as well. NA: not applicable; n.d.: not determined.**

| Sub-basin | Location | Size of area <30 m (km2) | Species incidence data set | | | | | Statistical estimations | | | Additional data sources | |
| --- | --- | --- | --- | --- | --- | --- | --- | --- | --- | --- | --- | --- |
| | | | Sample size (i.e. fishing/sampling occasions) | Species incidences[a] | SR$_{obs}$ | Singletons | Doubletons | IC (%) | SR$_{std}$ (with CI) | SR$_{est}$ (± SE)[b] | SR$_{obs}$ plus "species presence" observations from additional data sources | SR$_{obs}$ all countries and depths (HELCOM 2020)% |
| Bothnian Bay | Coastal | 5798 | 70 | 553 | 29 | 4 | 0 | 99.3 | $24^{28}_{25}$ | 35 ± 7 | 34 | 51 |
| | Offshore | 1376 | 4 | 23 | 8 | 1 | 1 | n.d. | n.d. | n.d. | 11 | |
| The Quark | Coastal | 1350 | 71 | 754 | 30 | 4 | 2 | 99.5 | $26^{28}_{24}$ | 34 ± 5 | 35 | 56 |
| | Offshore | 356 | NA | NA | NA | NA | NA | NA | NA | NA | NA | |
| Bothnian Sea | Coastal | 2986 | 194 | 2 222 | 42 | 2 | 0 | 99.9 | $37^{38}_{36}$ | 43 ± 2 | 50 | 74 |
| | Offshore | 918 | 17 | 87 | 24 | 5 | 3 | n.d. | n.d. | n.d. | 24 | |
| Åland Sea | Coastal | 240 | 31 | 394 | 32 | 6 | 3 | 98.5 * | $32^{35}_{28}$ * | 38 ± 6 | 45 | 71 |
| | Offshore | 28 | NA | NA | NA | NA | NA | NA | NA. | NA | NA | |
| N Baltic Proper | Coastal | 2156 | 77 | 1 046 | 42 | 6 | 3 | 99.4 | $36^{38}_{34}$ | 48 ± 6 | 56 | 67 |
| | Offshore | 218 | 4 | 11 | 4 | 0 | 2 | n.d. | n.d. | n.d. | 5 | |
| E Gotland Basin | Coastal | 936 | 13 | 94 | 25 | 8 | 5 | n.d. | n.d. | n.d. | 37 | 82 |
| | Offshore | 2148 | 6 | 45 | 14 | 4 | 3 | n.d. | n.d. | n.d. | 14 | |
| W Gotland Basin | Coastal | 5382 | 411 | 5 123 | 53 | 6 | 2 | 99.9 | $39^{40}_{38}$ | 62 ± 10 | 60 | 67 |
| | Offshore | 1525 | 4 | 20 | 7 | 1 | 2 | n.d. | n.d. | n.d. | 8 | |
| Bornholm Basin | Coastal | 972 | 68 | 837 | 46 | 5 | 1 | 99.4 | $42^{44}_{40}$ | 58 ± 17 | 59 | |





| | | | | | | | | | | | | |
|---|---|---|---|---|---|---|---|---|---|---|---|---|
| | Offshore | 2213 | 14 | 121 | 28 | 8 | 6 | n.d. | n.d. | n.d. | 31 | 104 |
| Arkona Basin | Coastal | 297 | 7 | 67 | 23 | 7 | 7 | n.d. | n.d. | n.d. | 37 | 110 |
| | Offshore | 1390 | 17 | 234 | 30 | 7 | 5 | n.d. | n.d. | n.d. | 30 | |
| The Sound | Coastal | 380 | 119 | 1 373 | 61 | 8 | 4 | 99.4 | $54_{51}^{57}$ | 69 ± 7 | 70 | 144 |
| | Offshore | NA | NA | NA | NA | NA | NA | NA | NA | NA | NA | |
| Kattegat | Coastal | 1906 | 353 | 6 012 | 101 | 11 | 8 | 99.8 | $78_{76}^{80}$ | 109 ± 6 | 114 | 178 |
| | Offshore | 1304 | 137 | 2 039 | 74 | 15 | 12 | 99.3 | $64_{59}^{68}$ | 83 ± 6 | 77 | |
| Skagerrak | Coastal | 1085 | 230 | 3 195 | 69 | 13 | 3 | 99.6 | $52_{50}^{53}$ | 90 ± 16 | 106 | NA |
| | Offshore | 80 | 5 | 65 | 30 | 14 | 6 | n.d. | n.d. | n.d. | 60 | |

[a] Sum of the number of species observed across all sampling occasions. Please note that this does not correspond to "entries" in Sect. 2.2, which is individual fish caught and determined to species.

[b] Considered a lower bound estimate (Chao et al., 2020)

### 3.3 Inventory completeness

The fish species IC in shallow coastal areas varied from 98.5% in the Åland Sea to 99.9% in the W Gotland Basin and the Bothnian Sea, based on analysis of data from the ten sub-basins with a sample size >25 fishings/samplings, suggesting that ca. 0.1-1.5% of statistically likely existing species remained undetected (Table 2). The species accumulation curves (SAC) show the SR$_{obs}$ at the conducted sample sizes, and SR estimated for hypothetical smaller and larger sample sizes, including 95% confidence intervals. According to these, the steepest increase of accumulated species occurred with the first ca. 20 samplings

in all sub-basins, and coastal fish SR was highest in the Kattegat, followed by the Skagerrak and The Sound, and lowest in the other seven sub-basins (i.e. confidence intervals not overlapping, Fig. 1a).

The SAC's also visualize differences in IC between sub-basins. For the three most saline sub-basins, Skagerrak, Kattegat and The Sound, the SACs were still clearly increasing with increasing sample size even when extrapolating to double the actual sample size. Hence, SR$_{est}$ for these sub-basins are more uncertain and more likely biased low than for sub-basins where the

curve flattened, illustrating a more complete inventory, e.g. W Gotland Basin and The Bothnian Sea (Fig. 1a). SR$_{est}$, estimated based on extrapolation of the information in the fish incidence database, were similar to SR$_{obs}$ if complementing the incidence data with records from the additional data sources (Table 2).



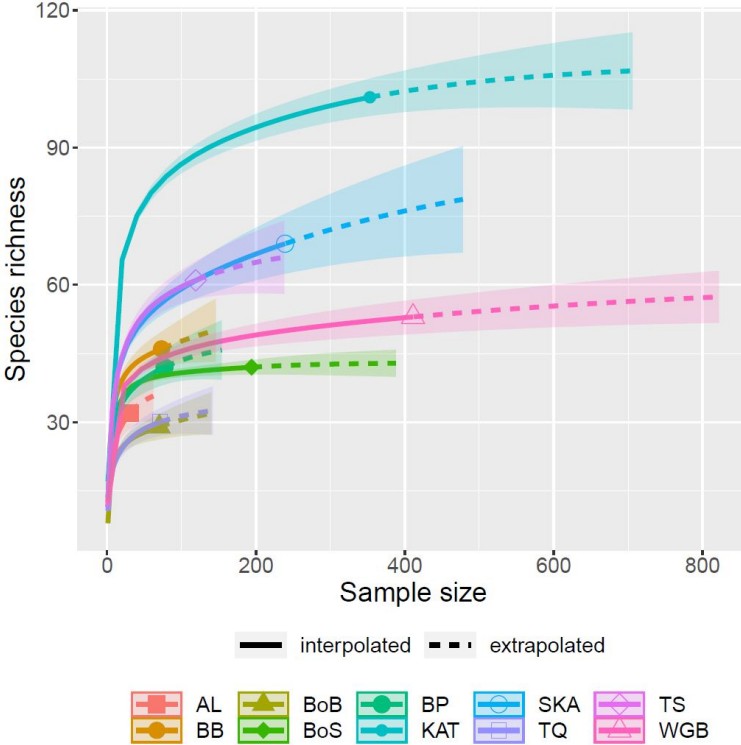

**Figure 1. Sample-size-based sampling curves with 95% confidence intervals (shaded areas), showing rarefaction/interpolation (solid) and extrapolation (dotted) line segments for (a) species richness (SR), (b) Shannon diversity (the effective number of frequent species in the assemblage, ShD) and (c) Simpson diversity (the effective number of very frequent species in the assemblage, SiD) of fish in coastal areas of the 10 analyzed sub-basins. The intersection points between solid and dotted lines represent the observed values. Legend acronyms are AL: Åland Sea, BB: Bornholm Basin, BoB: Bothnian Bay, BoS: Bothnian Sea, BP: N Baltic Proper, KAT: Kattegat, SKA: Skagerrak, TQ: The Quark, TS: The Sound and WGB: W Gotland Basin.**

For shallow offshore areas, only one sub-basin had enough data to conduct statistical rarefaction and extrapolation (i.e., Kattegat, Table 2). IC amounted to 99.3%, and also here $SR_{est}$ was similar to $SR_{obs}$ when incidence data and species presence information from additional sources were combined (Table 2). A comparison of $SR_{est}$ offshore areasuggests that, in Kattegat, fish SR is ca. 30% higher in coastal compared to offshore areas. A comparison based on $SR_{obs}$ when complementing the incidence data with additional data sources suggests ca. 50% higher SR in the coastal compared to offshore shallow Kattegat waters (Table 2).





**Table 3.** Shannon diversity (ShD) and Simpson diversity (SiD) for coastal and offshore areas. Calculated values are given for all sub-basins, and standardized (std) and estimated (est) values are given for the sub-basins with a sample size ≥ 25 fishings/samplingsShD gives the effective number of frequent species (the exponential of Shannon's entropy index), and SiD the effective number of highly frequent species (the inverse of Simpson's concentration index) in the assemblage (Chao et al. 2020, Chao et al. 2014). NA: not applicable; n.d.: not determined.

| Sub-basin | Area | Shannon Diversity | | | Simpson diversity | | |
|---|---|---|---|---|---|---|---|
| | | Calculated ShD | ShD$_{std}$ (with upper and lower confidence limits) | ShD$_{est}$ (± SE) | Calculated SiD | SiD$_{std}$ (with upper and lower confidence limits) | SiD$_{est}$ (± SE) |
| Bothnian | Coastal | 19 | $19^{20}_{18}$ | 20 ± 1 | 16 | $16^{17}_{15}$ | 16 ± 1 |
| Bay | Offshore | 8 | n.d. | n.d. | 7 | n.d. | n.d. |
| The Quark[a] | Coastal | 18 | $18^{18}_{17}$ | 18 ± 1 | 15 | $15^{16}_{14}$ | 15 ± 0.3 |
| Bothnian | Coastal | 28 | $27^{28}_{26}$ | 28 ± 0.4 | 23 | $23^{23}_{22}$ | 23 ± 0.4 |
| Sea | Offshore | 20 | n.d. | n.d. | 18 | n.d. | n.d. |
| Åland Sea[a] | Coastal | 21* | $22^{22}_{20}$ * | 22 ± 1 | 18* | $18^{19}_{17}$* | 18 ± 1 |
| N Baltic | Coastal | 26 | $25^{26}_{24}$ | 26 ± 1 | 22 | $22^{22}_{21}$ | 22 ± 0.4 |
| Proper | Offshore | 4 | n.d. | n.d. | 4 | n.d. | n.d. |
| E Gotland | Coastal | 18 | n.d. | n.d. | 15 | n.d. | n.d. |
| Basin | Offshore | 12 | n.d. | n.d. | 10 | n.d. | n.d. |
| W Gotland | Coastal | 31 | $29^{30}_{29}$ | 31 ± 0.2 | 26 | $25^{26}_{25}$ | 26 ± 0.3 |
| Basin | Offshore | 6 | n.d. | n.d. | 6 | n.d. | n.d. |
| Bornholm | Coastal | 32 | $31^{32}_{30}$ | 33 ± 1 | 27 | $26^{27}_{25}$ | 27 ± 1 |
| Basin | Offshore | 19 | n.d. | n.d. | 15 | n.d. | n.d. |
| Arkona | Coastal | 18 | n.d. | n.d. | 15 | n.d. | n.d. |
| Basin | Offshore | 21 | n.d. | n.d. | 18 | n.d. | n.d. |
| The Sound[a] | Coastal | 34 | $33^{34}_{32}$ | 35 ± 1 | 26 | $25^{26}_{24}$ | 26 ± 1 |
| Kattegat | Coastal | 51 | $49^{50}_{48}$ | 51 ± 1 | 38 | $37^{38}_{37}$ | 38 ± 0.4 |
| | Offshore | 32 | $31^{33}_{30}$ | 33 ± 1 | 24 | $24^{25}_{24}$ | 25 ± 0.4 |
| Skagerrak | Coastal | 33 | $32^{33}_{31}$ | 34 ± 1 | 25 | $25^{26}_{24}$ | 25 ± 0.3 |
| | Offshore | 25 | n.d. | n.d. | 22 | n.d. | n.d. |

[a] No offshore areas occur in these sub-basins.

### 3.4 Shannon and Simpson diversity

Rarefaction and extrapolation SACs carried out for Shannon diversity (ShD) show that the effective number of frequently recorded fish species was quite well captured by the samplings in all analyzed sub-basins, illustrated by SACs with small remaining slopes at extrapolated higher sample size. As for SR$_{obs}$, ShD was highest in Kattegat, while the remaining nine sub-basins clustered in two separate groups. The lowest ShD's were noted for the Åland Sea, The Quark and Bothnian Bay (Fig.



1b, Table 3). The effective number of highly frequent species, i.e. Simpson diversity (SiD), was also well captured in all sub-
basins, being highest in Kattegat, while SiD in the remaining sub-basins clustered in four groups (Fig. 1c, Table 3).

### 3.5 Standardized and estimated species richness

To compare coastal fish SR, ShD and SiD across sub-basins, we estimated their standardized values against the minimum observed IC in any of the sub-basins. This represented a standardization to the IC of the Arkona Basin data (98.5%; Tables 2 and 3). $SR_{std}$ was ca. three times higher in the relatively more saline Kattegat ($SR_{std} = 78$) compared to the least saline Bothnian
Bay ($SR_{std} = 24$), as also confirmed by comparing the respective $SR_{est}$ values (Table 2). The differences were smaller for ShD and SiD. For example, based on $SiD_{std}$ and $SiD_{est}$, the effective number of highly frequent species was ca. two times higher in coastal areas of the Kattegat compared to the Bothnian Bay (Table 3). This implies, as also seen from the SACs (Fig. 1), that the frequent and most frequent fish species were captured quite well by the samplings for all sub-basins, and that remaining uncertainties in differences across the salinity gradient is mostly due to uncertainty in the numbers of rare and very rare fish
species.

### 3.6 Relationships of SR with salinity and temperature

Species richness increased with increasing mean water salinity, which explained 37-55% of the variance in the data based on $SR_{obs}$, $ShD_{obs}$ and $SiD_{obs}$. Using the standardized or estimated values, i.e. values corrected for sample size, resulted in stronger correlations, i.e. higher explained variance (40-77%; Fig. 2, Table 4). $SR_{obs}$, $ShD_{obs}$ and $SiD_{obs}$ were not correlated with mean
water temperature, but, using the standardized and estimated values, correlations with temperature were also significant (explaining 48-77% of the variance; Fig. 2, Table 4). The slope estimates of the linear regressions differed more across observed, standardized and estimated values for SR than for ShD and SiD (Fig. 2, Table 4). In all cases, adding temperature as explanatory variable to the regression models with salinity as explanatory variable did not improve the model (all $P$>0.14).

### 3.7 Fish functional attributes

74% and 26% of the fish species recorded in shallow coastal areas were of marine and freshwater origin, respectively (based on the incidence data, i.e. $SR_{obs}$ of 92 vs. 33 species; Table S2). In the most saline sub-basins, i.e. Skagerrak and Kattegat, the $SR_{std}$ of marine fish species was seven to ten times higher than that of freshwater fish species. The $SR_{std}$ of marine vs. freshwater fish were rather similar in the central Baltic Sea, while in the northernmost and least saline sub-basins, i.e. Bothnian Sea, The Quark and Bothnian Bay, the $SR_{std}$ of freshwater fish species exceeded the $SR_{std}$ of marine fish species by two to three times.
In total, the marine fish $SR_{std}$ decreased by a factor of 8-11 along the salinity gradient, from 39 and 57 marine species ($SR_{std}$) in Skagerrak and Kattegat to 5 in the Bothnian Bay. Freshwater fish $SR_{std}$ increased by a factor of 2-4 along the same gradient (Fig. 3, Table S2). These distributional patterns of freshwater vs. marine fish species were also reflected by negative univariate




correlations of freshwater SR (obs, std and est) with salinity, and positive univariate correlations of marine SR with salinity (Fig. S1, Table 5).

**Table 4. Statistical indicators for the correlations between fish species richness (SR), Shannon Diversity (ShD) and Simpson Diversity (SiD), and salinity or annual mean water temperature in coastal areas of the studied sub-basins. The linear regressions were carried out separately for observed (*obs*), standardized (*std*) and estimated (*est*) values in each case. n.s.=not significant.**

| Response variable | | Salinity | | | Water temperature | | |
|---|---|---|---|---|---|---|---|
| | | Parameters (± SE) | Adjusted $R^2$ | *P*-value | Parameters (± SE) | Adjusted $R^2$ | *P*-value |
| SR | *obs* | $\log_{10}(y)=1.5$ (±0.1) + 0.014 (±0.004)*x | 0.55 | 0.004 | $\log_{10}(y)=1.2$ (±0.2) + 0.06 (±0.03)*x | 0.21 | n.s. (0.078) |
| | *std* | $\log_{10}(y)=1.45$ (±0.04) + 0.012 (±0.002)*x | 0.70 | 0.002 | $\log_{10}(y)=1.1$ (±0.1) + 0.07 (±0.01)*x | 0.77 | 0.001 |
| | *est* | $\log_{10}(y)=1.57$ (±0.04) + 0.014 (±0.002)*x | 0.77 | 0.001 | $\log_{10}(y)=1.2$ (±0.1) + 0.08 (±0.02)*x | 0.76 | 0.001 |
| ShD | *obs* | y=19.3 (±3.2) + 0.7 (±0.2)*x | 0.48 | 0.007 | y=5.0 (±10.6) + 3.1 (±1.4)*x | 0.25 | n.s. (0.055) |
| | *std* | y= 21.0 (±2.9) + 0.6 (±0.2)*x | 0.54 | 0.009 | y=2.3 (±7.6) + 3.7 (±1.0)*x | 0.57 | 0.007 |
| | *est* | y=21.7 (±3.1) + 0.7 (±0.2)*x | 0.55 | 0.009 | y=1.4 (±7.9) + 4.0 (±1.1)*x | 0.58 | 0.006 |
| SiD | *obs* | y=17.1 (±2.5) + 0.4 (±0.2)*x | 0.37 | 0.022 | y=7.8 (±7.8) + 2.0 (±1.0)*x | 0.19 | n.s. (0.087) |
| | *std* | y=18.4 (±2.3) + 0.4 (±0.1)*x | 0.41 | 0.027 | y=6.0 (±5.9) + 2.4 (±0.8)*x | 0.48 | 0.016 |
| | *est* | y=18.7 (±2.5) + 0.4 (±0.2)*x | 0.40 | 0.031 | y=5.3 (±6.2) + 2.6 (±0.8)*x | 0.48 | 0.016 |

Concerning habitat preference, half of the fish species in Swedish shallow coastal areas were classified as being coastal resident
species (CR; based on incidence data only, $SR_{obs}$: 63 species, Table S2). This group dominated coastal fish assemblages in all sub-basins, with CR $SR_{std}$ of 19-30 across sub-basins (Fig. 4, Table S2), and was not linearly related to salinity (Table 5). A similar result was noted for catadromous or anadromous fish species, with $SR_{std}$ between 2 and 6 in each sub-basin that was not related to salinity (Tables S2, 5). In the more saline sub-basins, fish species classified as marine visitors or as marine juvenile or seasonal migrants contributed significant numbers to the $SR_{std}$, while these species groups did not exist or
contributed only little to the $SR_{std}$ in the Baltic Sea region (Fig. 4, Table S2). Reflecting this pattern, the SR of marine migrating





or visiting fish species (i.e. MJ, MS and MV) was significantly positively related to salinity in most cases, with the strongest correlations for marine juvenile visitors (MJ; Fig. S2, Table 5).

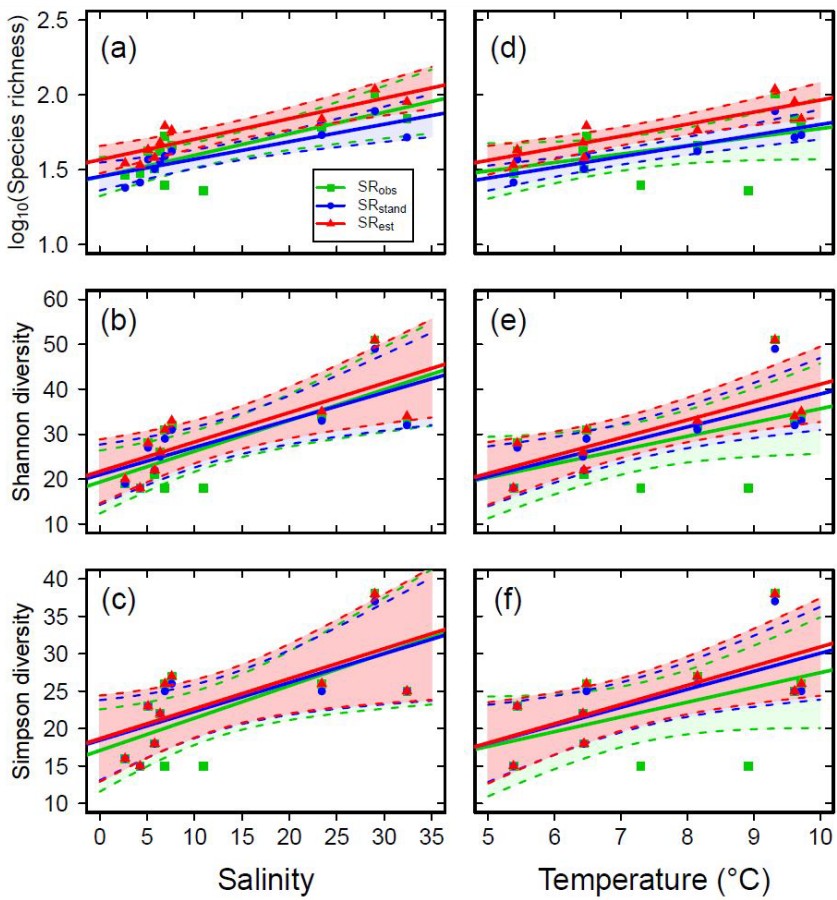

**Figure 2. Scatterplots of the fish species richness estimates against mean salinity (left column) and mean water temperature (right column), with total species richness (log$_{10}$-transformed; a and d), Shannon diversity (effective number of frequent species; b and e) and Simpson diversity (effective number of highly frequent species; c and f: Each plot shows the observed, standardized and estimated values, and, when significant ($P<0.05$), the linear regression lines (solid) and 95%-confidence intervals (shaded areas surrounded by dashed lines: The different lines and shaded confidence intervals are partly overlying each other within the panels in some cases, indicating very similar regression statistics. For regression equations and statistics, see Table 4.**

Concerning vertical distribution, benthic fish species (B) were important contributors to SR$_{std}$ in the sub-basins of higher salinity, but only few or no fish species belonged to this group in the less saline sub-basins (Fig. 5; Table S2). A similar, though less pronounced, distribution pattern was also found for demersal fish species (D). Accordingly, the SR of these groups were positively related to salinity in all cases (i.e. for SR$_{obs}$, SR$_{std}$ and SR$_{est}$, Fig. S3; Table 5). The SR of demersal-pelagic (DP) fish species varied between sub-basins with a SR$_{std}$ of 6-16, not related to salinity. A similar picture was found for pelagic fish species (P), where SR$_{std}$ varied between 5-12 across sub-basins (Fig. 5, Table S2) and was not related to salinity (Fig. S3, Table 5).





The two most common feeding groups observed in shallow coastal areas, across all sub-basins, were invertebrate and fish eating species (IF) as well as invertebrate feeders (I). The third-most represented feeding group was piscivorous fish species (Pi), followed by planktivorous and omnivorous species in lower and often similar $SR_{std}$ (Fig. 6, Table S2). $SR_{std}$ of Pi and IF

increased with increasing salinity, and $SR_{std}$ and $SR_{est}$ of I increased with increasing salinity (Fig. S4, Table 5).

**Table 5. Statistical relationships between observed (*obs*), standardized (*std*) and estimated (*est*) SR for fish functional attributes in Swedish shallow coastal areas and salinity. When YJ(y), the response variable was Yeo-Johnson transformed (Yeo and Johnson, 2000). n.s.=not significant.**

| Response variable | | | Parameters (± SE) | $R^2$ | *P*-value |
|---|---|---|---|---|---|
| Origin | Marine | *obs* | $\log_{10}(y)=1.0\ (\pm 0.1) + 0.03\ (\pm 0.01)*x$ | 0.74 | <0.001 |
| | | *std* | $\log_{10}(y)=0.9\ (\pm 0.1) + 0.03\ (\pm 0.01)*x$ | 0.71 | 0.001 |
| | | *est* | $\log_{10}(y)=1.1\ (\pm 0.1) + 0.03\ (\pm 0.01)*x$ | 0.66 | 0.003 |
| | Freshwater | *obs* | $y=22.0\ (\pm 3.0)\ -0.6\ (\pm 0.2)*x$ | 0.39 | 0.018 |
| | | *std* | $y=19.0\ (\pm 1.1) - 0.4\ (\pm 0.1)*x$ | 0.84 | <0.001 |
| | | *est* | $y=25.4\ (\pm 2.5) - 0.5\ (\pm 0.2)*x$ | 0.47 | 0.017 |
| Habitat preference | CR | *obs* | $y=25.3\ (\pm 4.6) + 0.1\ (\pm 0.3)*x$ | -0.08[a] | n.s. (0.661) |
| | | *std* | $y=26.5\ (\pm 2.0) - 0.1\ (\pm 0.1)*x$ | -0.04[a] | n.s. (0.538) |
| | | *est* | $y=32.5\ (\pm 3.0) + 0.1\ (\pm 0.2)*x$ | -0.10[a] | n.s. (0.665) |
| | CA | *obs* | $\log_{10}(y)=0.4\ (\pm 0.1) + 0.01\ (\pm 0.01)*x$ | -0.10[a] | n.s. (0.351) |
| | | *std* | $\log_{10}(y)=0.5\ (\pm 0.1) + 0.005\ (\pm 0.004)*x$ | -0.10[a] | n.s. (0.314) |
| | | *est* | $\log_{10}(y)=0.5\ (\pm 0.1) + 0.006\ (\pm 0.005)*x$ | -0.10[a] | n.s. (0.287) |
| | MJ | *obs* | $\log_{10}(y)=0.14\ (\pm 0.13) + 0.04\ (\pm 0.01)*x$ | 0.60 | 0.002 |
| | | *std* | $\log_{10}(y)=0.57\ (\pm 0.09) + 0.016\ (\pm 0.004)*x$ | 0.69 | 0.025 |
| | | *est* | $\log_{10}(y)=0.43\ (\pm 0.07) + 0.018\ (\pm 0.004)*x$ | 0.84 | 0.007 |
| | MS | *obs* | $y=0.7\ (\pm 0.6) + 0.3\ (\pm 0.04)*x$ | 0.79 | <0.001 |
| | | *std* | $y=2.4\ (\pm 1.0) + 0.2\ (\pm 0.1)*x$ | 0.67 | 0.029 |
| | | *est* | $y=3.0\ (\pm 1.4) + 0.2\ (\pm 0.1)*x$ | 0.55 | n.s. (0.056) |
| | MV | *obs* | $YJ(y)=-1.0\ (\pm 0.2) + 0.09\ (\pm 0.01)*x$ | 0.81 | <0.001 |
| | | *std* | $y=-3.4\ (\pm 7.1) + 0.7\ (\pm 0.3)*x$ | 0.65 | n.s. (0.123) |
| | | *est* | $y=-6.5\ (\pm 10.8) + 1.3\ (\pm 0.4)*x$ | 0.72 | n.s. (0.100) |
| Vertical distribution | B | *obs* | $YJ(y)=-1.0\ (\pm 0.2) + 0.09\ (\pm 0.01)*x$ | 0.76 | <0.001 |
| | | *std* | $\log_{10}(y)=0.3\ (\pm 0.1) + 0.031\ (\pm 0.004)*x$ | 0.91 | 0.001 |
| | | *est* | $\log_{10}(y)=0.4\ (\pm 0.1) + 0.039\ (\pm 0.004)*x$ | 0.95 | <0.001 |
| | D | *obs* | $\log_{10}(y)=1.0\ (\pm 0.1) + 0.02\ (\pm 0.01)*x$ | 0.44 | 0.011 |
| | | *std* | $\log_{10}(y)=0.97\ (\pm 0.04) + 0.013\ (\pm 0.013)*x$ | 0.70 | 0.002 |
| | | *est* | $\log_{10}(y)=1.1\ (\pm 0.1) + 0.014\ (\pm 0.004)*x$ | 0.54 | 0.010 |
| | DP | *obs* | $y=13.1\ (\pm 2.1) - 0.1\ (\pm 0.1)*x$ | -0.10[a] | n.s. (0.679) |
| | | *std* | $y=13.9\ (\pm 1.3) - 0.1\ \ (\pm 0.1)*x$ | 0.16 | n.s. (0.136) |





| | | | | | |
|---|---|---|---|---|---|
| | | *est* | y=15.5 (± 1.8) – 0.03 (± 0.11)*x | -0.12[a] | n.s. (0.792) |
| | | *obs* | $\log_{10}$(y)=0.8 (± 0.1) + 0.006 (± 0.004)*x | 0.07 | n.s. (0.203) |
| | P | *std* | $\log_{10}$(y)=0.80 (± 0.05) + 0.004 (± 0.003)*x | 0.06 | n.s. (0.243) |
| | | *est* | $\log_{10}$(y)=0.9 (± 0.1) + 0.008 (± 0.004)*x | 0.27 | n.s. (0.072) |
| | | *obs* | $\log_{10}$(y)=0.70 (± 0.06) + 0.018 (± 0.004)*x | 0.63 | 0.001 |
| | Pi | *std* | $\log_{10}$(y)=0.69 (± 0.06) + 0.014 (± 0.004)*x | 0.64 | 0.004 |
| | | *est* | $\log_{10}$(y)=0.76 (± 0.06) + 0.020 (± 0.004)*x | 0.77 | 0.001 |
| | | *obs* | y=8.7 (± 1.8) + 0.7 (± 0.1)*x | 0.75 | <0.001 |
| | IF | *std* | y=9.9 (± 1.1) + 0.4 (± 0.1)*x | 0.81 | <0.001 |
| | | *est* | y=10.7 (± 1.9) + 0.8 (± 0.1)*x | 0.84 | <0.001 |
| | | *obs* | $\log_{10}$(y)=0.8 (± 0.2) + 0.02 (± 0.01)*x | 0.16 | n.s. (0.108) |
| Feeding habit | I | *std* | $\log_{10}$(y)=0.85 (± 0.04) + 0.015 (± 0.003)*x | 0.79 | <0.001 |
| | | *est* | $\log_{10}$(y)=0.93 (± 0.04) + 0.018 (± 0.002)*x | 0.86 | <0.001 |
| | | *obs* | $\log_{10}$(y)=0.5 (± 0.1) + 0.007 (± 0.004)*x | 0.12 | n.s. (0.142) |
| | PL | *std* | $\log_{10}$(y)=0.4 (± 0.1) + 0.008 (± 0.003)*x | 0.36 | 0.041 |
| | | *est* | $\log_{10}$(y)=0.5 (± 0.1) + 0.007 (± 0.004)*x | 0.20 | n.s. (0.108) |
| | | *obs* | y=3.6 (± 0.9) – 0.1 (± 0.1)*x | -0.03[a] | n.s. (0.419) |
| | O | *std* | y=3.2 (± 0.5) + 0.1 (± 0.04)*x | 0.23 | n.s. (0.131) |
| | | *est* | y=3.5 (± 0.5) + 0.1 (± 0.04)*x | 0.33 | n.s. (0.080) |

[a]Adjusted $R^2$ can turn negative for multiple $R^2$ close to zero.




**Figure 3. Map of the study area covering the Baltic Sea and the Skagerrak, color-coded by mean salinity. Bar plots show standardized fish species richness for each of the ten analyzed sub-basins, separately for species of freshwater (F) and marine (M) origin. SR was**
**standardized across sub-basins to similar inventory completeness (Table S2: Black lines indicate the positions of the sub-basins, but the exact sampling sites were spread across the shallow areas of each of the sub-basins.**



**Figure 4. Map of the study area covering the Baltic Sea and the Skagerrak, color-coded by mean salinity. Bar plots show standardized**
**fish species richness for each of the ten analyzed sub-basins, separately by habitat preference category, as CR: coastal resident, CA: catadromous or anadromous migrants, MJ: marine juvenile migrants, MS: marine seasonal migrants and MV: marine visitors. SR was standardized across sub-basins to similar inventory completeness (Table S2: Black lines indicate the positions of the sub-basins, but the exact sampling sites were spread across the shallow areas of each of the sub-basins.**



**Figure 5. Map of the study area covering the Baltic Sea and the Skagerrak, color-coded by mean salinity. Bar plots show standardized fish species richness for each of the ten analyzed sub-basins, separately by vertical distribution category, with B: benthic, D: demersal, DP: demersal-pelagic and P: pelagic fish species. SR was standardized across sub-basins to similar inventory completeness (Table S2). Black lines indicate the positions of the sub-basins, but the exact sampling sites were spread across the shallow areas of each of the sub-basins.**
**Figure 6. Map of the study area covering the Baltic Sea and the Skagerrak, color-coded by mean salinity. Bar plots show standardized fish species richness for each of the ten analyzed sub-basins, separately by feeding category, with trophic level increasing from left to right, and PL: planktivores, O: omnivores, I: invertebrate eaters, IF: invertebrate and fish eaters and Pi: piscivores. SR was standardized across sub-basins to similar inventory completeness (Table S2). Black lines indicate the positions of the sub-basins, but the exact sampling sites were spread across the shallow areas of each of the sub-basins.**

## 4 Discussion

Data from species censuses have been called "probably the most basic data in ecology", as they are widely useful for example to define species ranges and biodiversity patterns, and support conservation efforts (Gaston & Blackburn, 2000). A limitation for the use of taxonomic inventory data for biodiversity purposes, however, is their completeness, i.e. the fraction of species in a given location that has been sampled (Mora et al., 2008). In this study, the coastal fish taxonomic IC was found to be ≥98.5% for the 10 analyzed sub-basins. This is high compared to a 2008 assessment of marine fish species census data





worldwide, where global IC averaged 79%, indicating that ca. 21% of fish species still remained to be described. Marine fish IC exceeded 80% in less than 2% of marine areas worldwide, and the highest IC of 92% was found for reef-associated species (Mora et al., 2008). Similarly, a 2012 global assessment concluded that ca. 77% of global marine fish SR were known to that

date. Consequently, the rate of new fish species descriptions continues to be high, with e.g. 1,577 new marine fish species globally described during the years 1999-2008 (Appeltans et al., 2012). A comparison between the estimated SR per sub-basin (statistical extrapolation of the fish incidence data) and the corresponding compilation of total observed species richness, i.e. also including species presence information from additional data sources than systematic sampling, yielded a mean ratio of $1.07 \pm 0.03$ (Table 2). This suggests that the overall observed fish species lists for Swedish shallow coastal areas are close to

complete for all analyzed sub-basins, and, in reverse, that the SR values estimated based on the fish incidence database ($SR_{est}$) were realistic.

The SR of frequent and very frequent species (i.e. Shannon and Simpson diversity, ShD and SiD) were generally well described by the sample sizes available to date in the studied sub-basins, with calculated ShD and SiD being similar to both standardized and estimated values (where effects of differing sample sizes are considered; Table 3). This indicates that the remaining

uncertainty in the fish $SR_{obs}$ is caused by a potential number of undetected rare species. This is a typical pattern, since well-known species are usually common and have large geographical ranges, whereas newly discovered species are usually (more) locally rare and geographically concentrated (Appeltans et al., 2012; Mora et al., 2008; Pimm et al., 2014).

The most recent check-list of Baltic Sea macrospecies, i.e. containing fish species reported across Baltic countries at both shallow and deeper water depths but excluding the Skagerrak, currently contains 242 fish species (HELCOM, 2020). In our

analyses of Swedish shallow coastal areas the total fish $SR_{obs}$ amounted to 144 (i.e., fish incidence data plus presence only data from additional data sources), also if Skagerrak is excluded. Comparing the sample-size corrected estimates of SR in coastal areas ($SR_{est}$) with HELCOM (2020) suggests that ca. 50-90% of the so far reported Baltic Sea fish species are currently found in Swedish shallow coastal areas, depending on sub-basin (data from 1975-2020).

Our study reinforces that $SR_{obs}$ is strongly dependent on IC, and that comparing $SR_{obs}$ of species assemblages without

accounting for this effect can lead to biased or even misleading conclusions (Chao & Chiu, 2016; Chao & Jost, 2015; Chao et al., 2020; Colwell & Coddington, 1994; Colwell et al., 2012; Gotelli & Colwell, 2001; Hill, 1973; Hsieh et al., 2016; Menegotto & Rangel, 2018; Mora et al., 2008; Pimm et al., 2014). Instead, when sample sizes are not uniform among sites or over time, $SR_{obs}$ need to be corrected for IC before valid conclusions can be made. However, such methods have so far only rarely been used for coastal and estuarine fish assemblages (Waugh et al., 2019).

Besides the effects of sample size, SR and IC might in this study also have been differentially influenced by variation in fishing methods, as the predominating methods differed across sub-basins. Multi-mesh gill nets dominated in seven of the statistically analyzed sub-basins, while trap nets and trawls dominated in the other three (Table S1). One "sample" represents a different effort depending on the gear used and method, and each gear has a specific selectivity and efficiency, which strictly does not allow for direct comparison (Bergström et al., 2013; Waugh et al., 2019). For example, at the Swedish west coast, gill nets

typically sample more species and individuals while fyke nets are more selective towards demersal and demersal-pelagic





species (Bergström et al., 2013). Merging the multi-gear data into one analysis may have caused a certain bias in this regard. However, we argue that our approach was feasible given that the fishing methods used in the different sub-basins are optimized for the locally prevailing conditions, i.e. aiming to sample the existing assemblages as completely as possible (Bergström et al., 2013), as additional data from relevant trawl surveys were also included, and considering the long time horizon of data

collection. Further supporting our approach, biodiversity metrics that were standardized against catch size revealed no consistent differences when comparing gill and fyke net samplings at the Swedish west coast (Bergström et al., 2013). Our assumption also appears justified given that $SR_{est}$ was similar to $SR_{obs}$ including additional data sources (i.e. incidence data plus presence observations, Table 2), giving confidence that the potentially introduced bias due to differing fishing gear and methods did not strongly influence the general patterns and results of this comparative and large-scale statistical analysis.

As anticipated based on earlier Baltic Sea studies on fish (e.g. Hiddink & Coleby, 2012; Ojaveer et al., 2010; Olsson et al., 2012) and other organism groups (e.g. Broman et al., 2019; Zettler et al., 2014), salinity was positively correlated with coastal fish SR (Table 4), with fish SR increasing ca. threefold across the ca. 10-fold salinity gradient (Table 2). That clear predominance of marine species in the most saline sub-basins compared to freshwater species in the inner parts of the Baltic Sea is in agreement with the fact that salinity functions as threshold or "ecological barrier" for the distribution of many

freshwater and marine species (Olenin & Leppäkoski, 1999; Vuorinen et al., 2015). It also corroborates patterns earlier reported for fish $SR_{obs}$ in three Baltic sub-basins (Hiddink & Coleby, 2012) and estuaries in general (Whitfield, 2015). The relatively small number of freshwater fish species incidences observed in the higher salinity sub-basins in our study (Fig. 3) likely stems from sampling close to freshwater tributaries, and reflects that many freshwater fish species can withstand extended exposure to certain salinity levels (<ca. 9) and tolerate brief exposure to higher salinities (>ca. 15; Peterson & Meador, 1994).

While temperature did not significantly correlate with observed SR, ShD or SiD, it was positively related with the standardized and estimated values (Table 4), which may indicate a temperature effect on fish biodiversity. Similarly, temperature has shown positive correlations with $SR_{obs}$ in North Atlantic demersal and benthopelagic fish assemblages (Gislason et al., 2020), and with fish $SR_{obs}$ in the coastal Norwegian Skagerrak (Lekve et al., 2002) as well as in estuaries worldwide (Vasconcelos et al., 2015), all being examples of the often found general pattern that broader-scale SR co-varies with climatic variables such as

temperature (Currie et al., 2004). However, given the clear relationship between salinity and the incidences of freshwater vs. marine fish species across the studied sub-basins (Fig. 3), we consider the studied salinity gradient to represent a case where the "physiological tolerance hypothesis" applies strongly, i.e. that SR in a particular area is limited by the number of species that can tolerate the local salinity conditions (Currie et al., 2004). In accordance, the regression models with salinity alone did not improve by adding temperature as additional explanatory variable. This conclusion is in agreement with observations from

estuaries that fish SR is influenced by the broader distributions and habitat preference patterns of marine and freshwater species that can colonize these areas (Vasconcelos et al., 2015).

In compiling data from the last nearly five decades we assumed that salinity changes during this time period (1975-2020) have been minor compared to the pronounced spatial salinity gradient. According to monitoring data, changes in salinity have been noted between ca. +3 psu in the Kattegat and ca. -1 psu in the Bothnian Sea during 1980-2015 (Ammar et al., 2021), which



can be considered small compared to the spatial salinity gradient ranging from 2 to 29. Moreover, fish populations often show
a lag of several years before biological changes following abiotic, environmental changes can be recorded (Daan et al., 2005).
Considering temporal patterns in SR and community composition, it was earlier reported that the observed fish SR increased
in Kattegat, Arkona Basin and the central Baltic during 2001-2008 (Hiddink & Coleby, 2012), and that the observed SR of
demersal fish increased in the Baltic Proper and the Bothnian Sea during ca. 1971-2013 (Törnroos et al., 2019). First-time

observations of known fish species in sub-basins where they were not previously caught have been related to increasing spring
temperatures (+3-6 °C during 1980-2015; (Ammar et al., 2021). Such potential temporal patterns were not analyzed here,
where we merged the fish incidence data across years to narrow down likely SR estimates for different sub-basins and focused
on large-scale spatial patterns.

Concerning habitat preferences, a higher proportion of resident fish species was found in the less saline sub-basins. This agrees

with with the observed predominance of freshwater species in these areas, while clearly migrating species are often of marine
origin (here classified as either marine juvenile migrants, marine seasonal migrants or marine visitors) and cannot tolerate low
salinity. The pattern is also in line with life strategies of fishes in marine coastal areas and European estuaries generally, having
an important role for ecological connectivity between open and coastal ecosystems (Franco et al., 2008). Concerning feeding
habits, relatively more species were higher trophic-level feeders in the more saline sub-basins, while a more even distribution

of feeding groups emerged towards less saline areas (Fig. 6, Table 5). Comparative analyses between coastal and offshore
areas could only be conducted for the Kattegatt, indicating a lower fish SR in the shallow offshore. This could be related to a
higher habitat complexity and more variable substrates in the coastal area, supporting more species (Bonsdorff, 2006).

Benthic and demersal fish SR decreased with decreasing salinity, corroborating previous results where demersal fish $SR_{obs}$
decreased from the saline Kattegat to the less saline northern Baltic Proper (Pecuchet et al., 2016). This pattern further

corresponds with that the observed SR of benthic meio- and macrofauna, which are the dominating prey for benthic fish, also
decreased with decreasing salinity in the Baltic Sea (Broman et al., 2019; Zettler et al., 2014). Taken together, and given that
most Baltic Sea fish species feed on benthic invertebrates during at least part of their life cycle (Snickars et al., 2015), these
patterns suggest that the strength of biological benthic-pelagic coupling through fish predation also likely differs along the
Baltic Sea salinity gradient.

**5 Conclusions**

Since fish SR and a number of functional attributes changed along the salinity gradient, respective changes in the coastal fish
communities may be foreseen if climate change further alters salinity conditions in the Baltic Sea. While the confidence in
future salinity projections remains low (HELCOM, 2021), recent ensemble simulations estimate that the two main drivers of
climate-related changes in salinity in the Baltic Sea region, increasing river runoff (leading to lower salinity) and sea level rise

(leading to higher salinity), approximately compensate each other, and may result in no net salinity changes (Meier et al.,
2021). Mean (depth-integrated) observed Baltic Sea salinity did not change during 1982-2016, however, vertical changes were



observed with freshening trends of the upper layer down to 40-50 m depth in most sub-basins, and increasing salinity below the halocline in the deep layer of some sub-basins (Liblik & Lips, 2019). Hence, if not considering potential phenotypical acclimation or genetic adaptation, an upper layer freshening would, based on the results from this and earlier studies (e.g.

Hiddink & Coleby, 2012; MacKenzie et al., 2007; Pecuchet et al., 2016), likely lead to less species-rich native fish communities in shallow coastal areas, where more and more marine species are excluded. Further, successful recovery of marine overfished species may become less probable while certain freshwater fish species may be favored (MacKenzie et al., 2007; Peterson & Meador, 1994). Indeed, marine fish species were negatively affected by a period of freshened conditions in the Baltic Sea during the ca. 1970-90s (Ojaveer & Kalejs, 2005). Benthic fish species, being mostly of marine origin, may be especially

vulnerable to freshening in the Baltic Sea region where their proportion in the fish assemblage is already relatively low to date. Besides salinity changes, fish SR and distribution may also be influenced by other climate-change related processes, including warming and resulting higher deep-water oxygen consumption rates, or changes in the Baltic Sea circulation (HELCOM, 2021; MacKenzie et al., 2007). Increasing water temperatures have already been linked to increased observed fish SR in the adjacent North Sea (Hiddink & Ter Hofstede, 2008), and in the Kattegat (Hiddink & Coleby, 2012). Further ecosystem-based

assessments are needed to obtain realistic predictions of the net effect of such ongoing environmental changes on future fish SR/community composition and on how they may interact with human activities such as fishing patterns, and with conservation needs for biodiversity management.

**Code and data availability**

The data used in this study is publicly available via the SLU Database for Coastal Fish – KUL

(https://www.slu.se/en/departments/aquatic-resources1/databases/database-for-coastal-fish-kul/), the SLU Swedish Species Information Centre (https://www.artportalen.se/), the SMHI SHARKweb (https://www.smhi.se/en/services/open-data/national-archive-for-oceanographic-data), the FishBase (https://fishbase.se/search.php) and the E.U. Copernicus Marine Service Information (CMEMS, 2021). The SLU Trawl Survey data (Department of Aquatic Resources) and the R code used for analysis and plotting is available from B. Koehler upon request.

**Author contribution**

LB, MK and BK designed the study. LB, ME and MK compiled the data. MK and LB assigned the functional characteristics to the fish species. BK analyzed the data. BK and ME visualized the data. All co-authors discussed and validated the results. BK prepared the manuscript with contributions from all co-authors.

The authors declare that they have no conflict of interest.





## Acknowledgements

The study was enabled by the WATERS project (Waterbody Assessment Tools for Ecological Reference conditions and status in Sweden), funded by the Swedish Environmental Protection Agency and the Swedish Agency for Water and Marine management, and by the Department of Aquatic Resources, Swedish University of Agricultural Sciences. Dr. Malin Werner
kindly contributed to the compilation of trawl data. The study is indebted to and was enabled by the Swedish national programs for fish, fisheries and marine environmental monitoring, hosted by SLU.

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
