# Peer review of "Species richness and functional attributes of fish assemblages across a large-scale salinity gradient in shallow coastal areas"

_Biogeosciences, 2021_

## Author Comment (AC2)

We would like to thank the reviewer for the thorough and constructive review of our manuscript "Species richness and functional attributes of fish assemblages across a large-scale salinity gradient in shallow coastal areas".

The assessment and comments are very helpful, and we agree with the suggested changes and needed clarifications/elaborations to improve the manuscript.

Please find our specific responses below, with suggested new or revised text parts in italic font.

**General comments**

Concerning the general comments, we agree it would benefit the clarity of the manuscript to remove the offshore data, given that only for one sub-basin there was enough data to conduct the statistical (rarefaction-extrapolation) analysis of fish species richness.

Regarding the specific comment concerning the title, i.e. that only coastal but not offshore areas were mentioned, this will be solved when the offshore data is excluded, following the reviewers suggestion.

We agree that the potential factors influencing fish SR should be mentioned already in the introduction. We therefore suggest to revise the following sentence in the introduction to include that aspect:

"*The species composition of fish in the Baltic Sea is regulated by salinity as well (Olsson et al., 2012; Pekcan-Hekim et al., 2016), even though other factors, such as temperature or habitat complexity, might also influence large-scale patterns of fish SR in estuaries (Vasconcelos et al., 2015; Schubert et al., 2011).*"

We further suggest to add two new discussion paragraphs on other regulating factors, specifically:

"*Besides salinity and temperature, which show a pronounced gradient over the large spatial scale of our study (Table 1) and were identified as likely main drivers here (Fig. 2), fish SR might also be influenced by other factors, such as human pressures. The cumulative pressure from human activities in the Baltic Sea, combining factors such as fishing, eutrophication and hazardous substances, is generally higher in the southern and south-western sub-basins. These sub-basins also show both relatively higher salinity and fish SR, compared to for example the northernmost sub-basins with lower cumulative human pressure, salinity and fish SR (Table 2; Korpinen et al., 2012; HELCOM 2018). This indicates that there is no negative relationship between cumulative human pressure and fish SR on the large spatial and temporal scales studied here. It does, however, not contradict well-documented influences that human pressures can have on fish concerning other aspects, or possibly for SR on smaller spatial scales (not studied here). For example, besides direct effects of fish extraction and habitat disturbance on fish species, human-induced depletion of larger predatory fish by Baltic Sea fisheries has likely contributed to an increase in coastal mesopredatory fish abundances (Eriksson et al., 2011), and eutrophication has been connected to increasing abundances of benthic feeding fish in the Baltic Sea (Snickars et al., 2015). Since the rarefaction-extrapolation analyses that we used here are based on species incidence frequencies (Chao et al., 2020), the statistical results could potentially be influenced by human pressures that alter these frequencies, even if fish SR in itself may not be affected. However, given that the rarefied and extrapolated SR (i.e. SR$_{std}$ and SR$_{est}$) are based on SC,*"

*where rare species are not influential, these statistics are rather robust against such effects (unless there would be severe changes in the incidence frequencies of common species). Another potential explanatory factor to consider is habitat complexity, related to e.g. diversity of substrate or habitat-forming macrophytes, which can increase aquatic biodiversity (Soukup et al., 2021). Differences in habitat complexity may play some role in the observed large-scale patterns in fish SR given that macroalgal SR increases with increasing salinity across the Baltic Sea, with a larger share of habitat-forming and perennial species in more marine waters (Middleboe et al., 1997, Schubert et al., 2011). Hence, the greater habitat complexity with increasing salinity may enhance fish SR, further reinforcing any salinity-induced distributional pattern.*"

We agree that further detail, explanation and clarification would benefit section 2.3, "Analysis of species richness data". In working on the foreseen revisions, we noted an unfortunate error in that the values of sample coverage had been incorrectly termed inventory completeness in the initially submitted manuscript (table and text). We will correct this, better explain the terms (following the reviewers suggestion) and include the values of inventory completeness (besides sample coverage) in Table 2.

To respond to this reviewers request to better elaborate and clarify the statistical terms and methods, we suggest to expand and revise section 2.3 to:

[revised manuscript text omitted]

We agree as well that a more detailed discussion of the fish functional characteristics and changes across the geographic gradient is warranted. We suggest to expand and revise the respective discussion section, starting L449, to:

*"Our study also revealed changes in fish SR for different functional groups across the studied salinity gradient. As the different functional groups represent variability among species in e.g. use of resources or level of connectivity with other areas, this may also translate to taxonomic-driven differences in coastal ecosystem functioning across the different sub-basins (Elliott et al., 2007; Franco et al., 2008). Clearly migrating fish species are typically of marine origin (here classified as marine juvenile migrants, marine seasonal migrants or marine visitors) and cannot tolerate low salinity, explaining their predominance at higher salinities (Fig. 4), and in agreement with known patterns in European estuaries in general (Elliott & Dewailly, 1995; Franco et al., 2008). This pattern of marine fish species temporarily using coastal areas may be related to comparatively higher prey densities and to food types not encountered in marine areas, as well as to typically more turbid waters providing better protection from predators (Franco et al., 2008). Moreover, the high migratory fish SR at higher salinity is likely relevant for the ecological connectivity between ecosystems, e.g. by transport of local "coastal" production to open sea and vice versa (Franco et al., 2008), and emphasizes the important role of higher salinity coastal areas as nursery grounds, migration routes and refuge areas for marine fish species (Elliott et al., 2007). Connectivity is also maintained in the less saline sub-basins, though the concerned functional groups are represented by only a few species (Fig. 1, S2; Berkström et al., 2021).*

*Benthic and demersal fish SR also decreased with decreasing salinity, corroborating previous results where demersal fish $SR_{obs}$ decreased from the saline Kattegat to the less saline northern Baltic Proper (Pecuchet et al., 2016), and in accordance with high benthic preference of marine fish species in European estuaries (Elliott & Dewailly, 1995; Franco et al., 2008). This pattern further corresponds with that the observed SR of benthic meio- and macrofauna, which are generally the dominating prey for benthic and demersal fish, also decreased with decreasing salinity in the Baltic Sea (Broman et al., 2019; Zettler et al., 2014). Taken together, these patterns suggest that the process of benthic-pelagic coupling through fish predation likely involves a lower number of species links, or functional redundancy, towards lower salinity sub-basins. Concerning feeding habits, the general composition of feeding guilds noted in the higher-salinity sub-basins was similar to that reported on a larger European scale (Elliott & Dewailly, 1995). Also our findings of higher*

*piscivorous fish SR in the more saline sub-basins (Fig. 6) and the pattern showing a low proportion of omnivorous fish that was unrelated to salinity levels was in agreement with findings from European estuaries (Franco et al., 2008). In summary, the found differences in functional traits of fish along the salinity gradient were largely related to the respective changes in the predominating fish origin, i.e. freshwater vs. marine species.*"

**Specific comments**
We thank the reviewer for pointing out the needed technical corrections and suggestions for improving tables, and will follow all of them.

Reflecting the revisions conducted to the manuscript we further suggest to slightly revise/adapt the Abstract to:

"*Coastal ecosystems are biologically productive and their diversity underlies various ecosystem services to humans. However, large-scale species richness (SR) and its regulating factors remain uncertain for many organism groups, owing not least to the fact that observed SR ($SR_{obs}$) depends on sample size and inventory completeness (IC). We estimated changes in SR across a natural geographical gradient using statistical rarefaction and extrapolation methods, based on a large fish species incidence dataset compiled for shallow coastal areas (<30 m depth) from Swedish fish survey databases. The data covered a ca. 1,300 km north-south distance and a 12-fold salinity gradient along sub-basins of the Baltic Sea plus Skagerrak and, depending on sub-basin, 4 to 47 years of samplings during 1975–2021. Total fish $SR_{obs}$ was 144, and the observed fish species were of 74% marine and 26% freshwater origin. In the 10 sub-basins with sufficient data for further analysis, IC ranged from 77–98%, implying that ca. 2–23% of likely existing fish species remained undetected. Sample coverage ranged from 98.5–99.9%, suggesting that the undetected species represented <1.5% of incidences across the sub-basins, i.e. highly rare species. To compare between sub-basins, we calculated standardized SR ($SR_{std}$) and estimated SR ($SR_{est}$). Sub-basin specific $SR_{est}$ varied between 35 ± 7 (SE) and 109 ± 6 fish species, being ca. three times higher in the most saline (salinity 29–32) compared to the least saline sub-basins (salinity <3). Additional information on functional attributes showed that differences with decreasing salinity particularly reflected a decreasing SR of benthic and demersal fish, piscivorous and invertebrate-eating fish, and marine migratory fish. We conclude that, if climate change continues causing an upper-layer freshening of the Baltic Sea, this may influence the SR, community composition and functional characteristics of fish, which in turn may affect ecosystem processes such as benthic-pelagic coupling and connectivity between coastal and open sea areas.*".

We thank the reviewer and Editors for their efforts with this work, and are looking forward to hear from you about our manuscript.

Yours sincerely,

Birgit Koehler and co-authors

---

## Author Comment (AC3)

We would like to thank the reviewer for the thorough and constructive review of our manuscript "Species richness and functional attributes of fish assemblages across a large-scale salinity gradient in shallow coastal areas".

The assessment and comments are very helpful, and we overall agree with the suggested changes and needed clarifications/elaborations to improve the manuscript.

Please find our specific responses below, with suggested new or revised text parts in italic font.

1) Manuscript structure, tables and figures

The reviewer suggests the salinity map to be shown only once in the main manuscript. We agree on the point made, and, to accomplish this while maintaining the necessary information in the main manuscript, we suggest to move Figures 4-6 into the Supplement and instead lift the previous Figures S2 and S3 into the manuscript. These show the relationships between functional groups and salinity as bi-plots (with regression lines when significant). At the same time, we would like to complete these figures, that would now be shown in the main manuscript, by including all functional groups, where we previously (i.e. in the Supplement) had only included a subset of groups. This will facilitate complete overview over the found relationships when most of the current maps should be moved to the Supplement.

By these suggested changes to the Figures, their number in the manuscript would be reduced from previously six to five. The acronyms of the sub-basins, which are given in the bar-plots of Figure 1, will be defined in the respective figure legends.

We will also follow the reviewers suggestions and place a larger-scale map for reference into the maps showing the Baltic Sea with functional patterns.

Further, we suggest to move Table 3, which gives the observed, standardized and estimated Shannon and Simpson Diversity, as well as Table 4, which gives regression information for relationships between Shannon and Simpson Diversity with salinity and temperature, from the manuscript into the Supplement. These revisions will accommodate the comment that the three estimates (std, obs, est) are, for the more common species, very similar. Figure 2, which would remain in the manuscript, will still show both the values (referring to previous Table 3) and relationships to salinity (referring to previous Table 4). By these revisions, the total number of Tables in the manuscript would be reduced from previously five to three. Further, the tables as such will become considerably simpler following the exclusion of the offshore data from the manuscript, as suggested by the reviewer.

We would suggest to keep all three estimates (i.e. std, obs and est) in Table 2 for species richness, as these differ more than for ShD and SiD (since the rare species are included). One of our central messages, i.e. that fish SR decreased about three-fold from the highest to the lowest salinity sub-basin, is based on $SR_{std}$. It is also relevant to compare $SR_{est}$ with e.g. total observed SR (i.e. based on incidence data plus presence observations). For these reasons we also suggest to keep Table 2 in form of a Table, such that the precise values (and error estimates) can be directly compared. To accommodate the reviewers comment, we suggest to better explain the three different estimates in the revised Methods section, section 2.3, "Analysis of species richness data" (see below for details). In short, the purpose of the standardized estimate is to enable accurate comparisons among areas, in spite of differing sample sizes and inventory completeness. The purpose of the estimated SR is to extrapolate to a likely number of species in one area, if inventory was continued. The complementarity of

the different estimates is, for example, discussed in the last two sentences of the first discussion paragraph, and in the second paragraph of the discussion.

2) Coastal resident vs. migrating species

Since the marine migrating and visiting species are part of the coastal fish assemblage during certain parts of the year, and depend on coastal areas during certain parts of their life cycle, we included them in the statistical analyses. The mix of resident and mobile fish species indicates the dynamic nature of certain coastal areas, and their connectivity with the open sea. We agree with the reviewer that inclusion or exclusion of marine migrating and visiting fish species in the analyses does affect how the inventory completeness (IC) turns out, but find that including them results in a more complete representation of SR. To clarify this, we suggest to add a note in the respective Methods section (Sect. 2.3) which details that both resident and migrating/visiting species were included during calculation of IC, and to add a discussion around the implications of this in the discussion section, as:

*"The comparatively low IC in Skagerrak (77%, Table 2) can be related to a, in this context, rather small sample size for a relatively high number of marine migrating/visiting species (which are only more rarely present; Table S5), indicating that the fish species composition in this sub-basin is more "dynamic"."*

We further agree with the reviewer that a lower or higher proportion of marine migrating and visiting species would influence the estimate of IC, and that changes in this proportion over time would affect the IC estimate. Adding such a temporal perspective would be interesting to follow up in a future study, but was not included in this large-scale spatial study. In the current study, monitoring data from several years were merged over time in order to obtain as accurate as possible spatial comparisons within the limits of available data. It is important to note, though, that the sample coverage (SC) in our study, i.e. the inventory completeness of the more common species, is not strongly affected by the incidence frequency of rare species, and was very similar across sub-basins (Table 2). Since $SR_{std}$ and $SR_{est}$ depend on SC they are rather robust against changes in the proportion of (more rarely present) migrating/visiting vs. resident species, and hence against including or excluding migrating and visiting species. We suggest to include discussion around this aspect, specifically:

*"Since the rarefaction-extrapolation analyses that we used here are based on species incidence frequencies (Chao et al., 2020), the statistical results could potentially be influenced by human pressures that alter these frequencies, even if fish SR in itself may not be affected. However, given that the rarefied and extrapolated SR (i.e. $SR_{std}$ and $SR_{est}$) are based on SC, where rare species are not influential, these statistics are rather robust against such effects (unless there would be severe changes in the incidence frequencies of common species)."*

3) Offshore data

We agree that it will benefit the general clarity of manuscript to not include the offshore data, particularly given that only one sub-basin had enough data to conduct the statistical (rarefaction-extrapolation) analysis of fish SR.
Indeed, as the reviewer assumed, data from the BITS survey carried out in the Baltic Sea were not included due to a lack of data for shallower depths (i.e. <30 m).

4) Discussion on other drivers

We agree with the reviewer, and suggest to include a new discussion paragraph on the potential influence of other drivers, such as human pressures and habitat complexity, on fish SR and on our statistical results, specifically:

*"Besides salinity and temperature, which show a pronounced gradient over the large spatial scale of our study (Table 1) and were identified as likely main drivers here (Fig. 2), fish SR might also be influenced by other factors, such as human pressures. The cumulative pressure from human activities in the Baltic Sea, combining factors such as fishing, eutrophication and hazardous substances, is generally higher in the southern and south-western sub-basins, which also have both relatively higher salinity and fish SR, compared to for example the northernmost sub-basins with lower cumulative human pressure, salinity and fish SR (Table 2; Korpinen et al., 2012; HELCOM 2018). This indicates that there is no negative relationship between cumulative human pressure and fish SR on the large spatial and temporal scales studied here. It does, however, not contradict well-documented influences that human pressures can have on fish concerning other aspects, or possibly for SR on smaller spatial scales (not studied here). For example, besides direct effects of fish extraction and habitat disturbance on fish species, human-induced depletion of larger predatory fish by Baltic Sea fisheries has likely contributed to an increase in coastal mesopredatory fish abundances (Eriksson et al., 2011), and eutrophication has been connected to increasing abundances of benthic feeding fish in the Baltic Sea (Snickars et al., 2015). Since the rarefaction-extrapolation analyses that we used here are based on species incidence frequencies (Chao et al., 2020), the statistical results could potentially be influenced by human pressures that alter these frequencies, even if fish SR in itself may not be affected. However, given that the rarefied and extrapolated SR (i.e. SRstd and SRest) are based on SC, where rare species are not influential, these statistics are rather robust against such effects (unless there would be severe changes in the incidence frequencies of common species). Another potential explanatory factor to consider is habitat complexity, related to e.g. diversity of substrate or habitat-forming macrophytes, which can increase aquatic biodiversity (Soukup et al., 2021). Differences in habitat complexity may play some role in the observed large-scale patterns in fish SR given that macroalgal SR increases with increasing salinity across the Baltic Sea, with a larger share of habitat-forming and perennial species in more marine waters (Middleboe et al., 1997, Schubert et al., 2011). Hence, the greater habitat complexity with increasing salinity may enhance fish SR, further reinforcing any salinity-induced distributional pattern."*

5) Discussion on functional attributes (traits)

We agree that the fish functional characteristics and their changes along the salinity gradient warrant more detailed discussion, that the statement on benthic-pelagic coupling needs to be clarified, and that it would be interesting to couple the discussion to ecosystem processes. To meet these comments, and as an overall revision in response to both reviewers' comments on this discussion section, we suggest to revise and expand the respective discussion paragraphs to:

*"Our study also revealed changes in fish SR for different functional groups across the studied salinity gradient. As the different functional groups represent variability among species in e.g. use of resources or level of connectivity with other areas, this may also translate to taxonomic-driven differences in coastal ecosystem functioning across the different sub-basins (Elliott et al., 2007; Franco et al., 2008). Clearly migrating fish species are typically of marine origin (here classified as marine juvenile migrants, marine seasonal migrants or marine visitors) and cannot tolerate low salinity, explaining their predominance at higher salinities (Fig. 4), and in agreement with known patterns in European estuaries in general (Elliott & Dewailly, 1995; Franco et al., 2008). This pattern of marine fish species temporarily using coastal areas may be related to comparatively higher prey densities and to food types not encountered in marine areas, as well as to typically more turbid waters*

*providing better protection from predators (Franco et al., 2008). Moreover, the high migratory fish SR at higher salinity is likely relevant for the ecological connectivity between ecosystems, e.g. by transport of local "coastal" production to open sea and vice versa (Franco et al., 2008), and emphasizes the important role of higher salinity coastal areas as nursery grounds, migration routes and refuge areas for marine fish species (Elliott et al., 2007). Connectivity is also maintained in the less saline sub-basins, though the concerned functional groups are represented by only a few species (Fig. 1, S2; Berkström et al., 2021).*

*Benthic and demersal fish SR also decreased with decreasing salinity, corroborating previous results where demersal fish SRobs decreased from the saline Kattegat to the less saline northern Baltic Proper (Pecuchet et al., 2016), and in accordance with high benthic preference of marine fish species in European estuaries (Elliott & Dewailly, 1995; Franco et al., 2008). This pattern further corresponds with that the observed SR of benthic meio- and macrofauna, which are generally the dominating prey for benthic and demersal fish, also decreased with decreasing salinity in the Baltic Sea (Broman et al., 2019; Zettler et al., 2014). Taken together, these patterns suggest that the process of benthic-pelagic coupling through fish predation likely involves a lower number of species links, or functional redundancy, towards lower salinity sub-basins. Concerning feeding habits, the general composition of feeding guilds noted in the higher-salinity sub-basins was similar to that reported on a larger European scale (Elliott & Dewailly, 1995). Also our findings of higher piscivorous fish SR in the more saline sub-basins (Fig. 6) and the pattern showing a low proportion of omnivorous fish that was unrelated to salinity levels was in agreement with findings from European estuaries (Franco et al., 2008). In summary, the found differences in functional traits of fish along the salinity gradient were largely related to the respective changes in the predominating fish origin, i.e. freshwater vs. marine species."*

Concerning the other comments:
- L32: We refer to species richness, and suggest to clarify the statement accordingly by revising to: "However, threats to coastal biodiversity from e.g. overfishing, habitat loss, pollution, eutrophication and climate change are many and profound (Duncan et al., 2015; Griffiths et al., 2017; Pan et al., 2013), and the number of species occurring in coastal habitats often remains uncertain (Appeltans et al., 2012)."
- L49: We suggest to add "on average" to accommodate the fact that certain regions experience less intense water cycling, while the general trend is an intensified hydrological cycle.
- L78: This comment does not anymore apply since offshore areas will be, upon suggestion from the reviewer, removed from the manuscript (see above).
- L116: In separated ecosystems, such as islands or lakes, SR usually increases with area ("species-area relationship"). However, such separation is not the case for our coastal sub-basins here. While the studied sub-basins are hydrographically distinct, with water exchange being separated to a certain degree by shallow sounds or sills, they are still connected. Based on this reviewers comment we realised that giving the size of the shallow coastal areas in table and Methods may cause confusion in this regard. We therefore suggest to remove these values, which are also not used further in the study, leading to simplification of tables (as suggested would be useful by the reviewer).
- L128: We suggest to revise to "*Since each gear has a specific selectivity and efficiency this may have introduced a bias in the dataset. However, a comparison of*

*data from gill and fyke net samplings at the Swedish west coast did not reveal consistent differences in biodiversity metrics, and the statistical approaches were chosen to minimize this potential bias when comparing SR among sub-basins (Bergström et al., 2013, Chao et al., 2020; see also search of additional data sources below, and Sects. 2.3 and 4)."*

- L139: We suggest to revise this part to better explain the motivation for the chosen cutoff. It was essentially based on that we found that sub-basins with less than one hundred fish species incidences had too little data for statistical analysis:
*"Data on observed SR, $SR_{obs}$, is available for all 12 sub-basins (Table 2), and was based on between 4 and 47 years of data, depending on sub-basin (Table S2). However, subsequent statistical analyses and comparisons were conducted only for the 10 sub-basins containing data from at least 30 sampling/fishing occasions, corresponding to several hundred fish incidences (i.e. Bothnian Bay, the Quark, Bothnian Sea, Åland Sea, N Baltic Proper, W Gotland Basin, Bornholm Basin, the Sound, Kattegat, and Skagerrak). This dataset is hereafter referred to as "raw data", and contained in total 160,453 entries (i.e. fish individuals caught and determined to species) from 1,638 sampling/fishing occasions at 4,571 unique locations. E Gotland Basin and Arkona Basin were not statistically analysed since we considered these sub-basins too under-sampled, with only 13 and 7 samplings, respectively, from 9 and 4 different years, and less than 100 species incidences in total (Tables 2, S2)."*
While we extracted all available data from the database covering nearly five decades (1975-2021), as stated in the manuscript, only one sub-basin had annual samplings during all those years, while most sub-basins had data from 17+ years (with a few having less years, i.e. 4 and 9 years, and being excluded from statistical analysis based on that, see above). We realized this aspect would need to be clarified in the manuscript, and suggest to edit the text accordingly, and to add a new Supplementary Table detailing for which years fishing data was available, per sub-basin.

- L110-147: The incidence data is official, quality-controlled survey data for which species are correctly identified by taxonomic specialists. As an additional double-check of uncommon species we used the HELCOM list of macro-species in the Baltic Sea. Concerning the "observation databases", where e.g. citizens can report species observations and which hence are less reliable, we did a careful cross-check where unreasonable occurrences were considered falsely identified, and discarded (see L139/140). We will elaborate to clarify this in the respective text pieces.

- L167: The comment made us realise that this text part could be confusing since the rarefaction-extrapolation method used is called "Chao Richness method", and a function available in R has the same name although it includes several different calculations. Please see below for how we suggest to revise this section, to accommodate this and other comments on it.

- L170: We suggest to clarify that the observed values were standardized to the minimum SC. The purpose of $SR_{std}$ is to allow for accurate comparisons between sub-basins, given that all standardized values give SR for the same SC, hence representing an estimate which is not biased by how completely the compared areas were sampled. During revision, we noted an unfortunate mistake in that "sample coverage" (SC) was erroneously called "inventory completeness" (IC). This is now corrected, and better explained in the methods section 2.3. We agree with the reviewer that SC was high and very similar across sub-basins, varying between 98.5-99.9%, and hence in the case of this study did not strongly influence the obtained values. However, the correction is still needed in order to allow for accurate, unbiased comparisons between sub-basins,

and we therefore find it important to keep the standardization in the manuscript rather than moving it to the Supplement. However, we understand the need of presenting the data and results more clearly, especially in the Tables. We suggest that this aim will be achieved by excluding the offshore data, as suggested by both reviewers and making text and tables considerably simpler, and by reducing the number of tables and figures into (please see above for details).

To further accommodate this comment and the comment above we also suggest a major rewrite of the method section 2.3, to:

"*The raw data was first summarized to a dataset of unique fish species caught per fishing/sampling occasion in presence/absence format, and then further aggregated to an incidence frequency format, giving the observed total incidence of each species over the number of fishing/sampling occasions. This dataset is referred to as "fish incidence database". Each unique combination of a fishing/sampling location per date was defined as one sampling unit, and these were summed per sub-basin to obtain the sample sizes. Subsequently, incidence-based Hill diversity numbers of three "orders", which differ in their propensity to include or exclude relatively rarer species (Hill, 1973), were calculated to quantify the species diversity of each assemblage, i.e. 1) species richness (SR), which counts all species equally irrespective their incidence frequency, 2) Shannon diversity (ShD), which considers the incidence frequency and can be interpreted as the effective number of frequent species, and 3) Simpson diversity (SiD), which can be interpreted as the effective number of highly frequent species (Chao et al., 2014; Chao et al., 2020). Calculations were performed using the R package iNEXT (Chao et al., 2020; Hsieh et al., 2016), and the values are hereafter referred to as observed SR, ShD and SiD, respectively. It should be noted that, using these methods, Shannon and Simpson diversity are expressed in terms of richness, i.e. number of species, which differs from other known formats. Specifically, ShD is the exponential of Shannon's entropy index, and SiD is the inverse of Simpson's concentration index (Chao et al., 2014).*

*$SR_{obs}$ is highly dependent on "sample completeness" (Colwell & Coddington, 1994; Hill, 1973) and typically underestimates the "true" SR due to undetected species, an aspect referred to as under-sampling, sampling bias or sampling problem (Chao et al., 2014; Chao & Jost, 2015; Menegotto & Rangel, 2018). Similar to Hill numbers, "sample completeness" can be calculated for different "orders" (Chao et al., 2020). The zero-order sample completeness is hereafter referred to as inventory completeness (IC). It is calculated as the ratio of $SR_{obs}$ to the estimated "true" SR (i.e. observed plus undetected SR, see "estimated SR" below), hence giving the proportion of detected species without considering the species incidence frequencies. We calculated IC for the data merged over time, and including both resident and migrating/visiting fish species. The first-order sample completeness, hereafter referred to as "sample coverage" (SC), is a measure where species are weighted by their detection probabilities, giving the proportion of incidences detected from the estimated "true" incidences (Chao et al., 2020).*

*To correct for the effect of differing sample completeness on $SR_{obs}$, and allow accurate, unbiased comparisons between sub-basins, we used a coverage-based rarefaction and extrapolation method implemented for incidence data in the R package iNEXT (Chao et al., 2014, 2020; Hsieh et al., 2016). A coverage-based method was chosen because more traditional sample size-based corrections can introduce a systematic bias, since the number of samples needed to fully characterize a community depends on its SR (Chao & Jost, 2012). For each sub-basin, we obtained 1) the rarefied SR, ShD and SiD, which were standardized to the minimum observed*

*SC across all included sub-basins ( hereafter referred to as standardized values, i.e. $SR_{std}$, $ShD_{std}$ and $SiD_{std}$), and 2) the fish SR extrapolated to twice the actual sample size  (hereafter referred to as estimated values, i.e. $SR_{est}$, $ShD_{est}$ and $SiD_{est}$; Chao et al., 2014, 2020; Hsieh et al., 2016). Similar analyses were also conducted for SR of fish with different functional attributes (see Sect. 2.4). All calculations were conducted using R version 4.0.4 (R Core Team, 2021).".*

- L222: This comment does not apply any more following omission of the offshore data, as suggested by both reviewers.
- L237: This part should be changed, because we realized during revision that we had by mistake interchanged the terms inventory completeness and sample coverage. Now, inventory completeness shows a larger variation, and we suggest to refer to Table 2 rather than translating the percentages into species numbers.
- L250: Extrapolation is recommended up to maximally twice the actual sample size for SR (Chao et al., 2020). We suggest to include this information in the respective methods section, and in the legend of Figure 1.
- Table 3. We suggest to move this table to the Supplement (please also see above). A graphic illustration of the trends across the salinity gradient is available in Figure 2, and, for ShD and SiD, the observed values are very similar to the standardized and estimated values, given that the missing species represent rare species. Hence, for study sites with high SC, the observed ShD and SiD are already largely unbiased estimates. To further accommodate the reviewers general comment that the manuscript may contain too many tables and figures we suggest to also move Table 4 to the Supplement. Table 4 contains the statistical information on regressions between salinity or temperature with SR, ShD and SiD. Since these regressions are already graphically shown in Figure 2 it seems justified to move the statistical details on their relationships to the Supplement.
- Table 5. We will include information about the number of observations in Table 5, and in the legend of Table 4.
- Figures 4-6. We agree it is redundant to repeat the salinity map repeatedly in these figures. We therefore suggest to move figures 4, 5 and 6 into the Supplement, and instead lift the previous figures S2 and S3 into the manuscript. These figures directly shows the relationships between functional groups (on habitat use and vertical distribution) and salinity.
- L372: Mora et al., 2008 applied rarefaction-extrapolation methods similar to the one used in our study, but used different statistical models. Appeltans et al. 2012 based their estimate of inventory completeness on a statistical model based on historical rates of species description. While the methods differ more or less from the one we used the estimated property is the same, i.e. the inventory completeness as the proportion of observed to total species, which motivates our comparison in spite of methodological variation.
- L379: We suggest to rewrite this sentence to clarify which ratio we mean, and to move it to the Results (end of Sect. 3.5), but to take it up as discussion point in this place. We also suggest a rewrite to "calculated based on data presented in Table 2", to make clear that the ratio is not given in the Table, but that the values needed to calculate it are found in Table 2.
- L394: We understand that this statement may be confusing. We suggest a rewrite to: *"Our study reinforces that $SR_{obs}$ is strongly dependent on sample coverage, as relatively rare species are more likely to be missed at lower sample size/sample coverage, and that comparisons of $SR_{obs}$ of in species assemblages without accounting for this effect can lead to biased or even misleading conclusions.".*

- L400-414: We suggest to add some more information on this aspect already in the respective Methods section, section 2.2, specifically: "*Since each gear has a specific selectivity and efficiency this may have introduced a bias in the dataset. However, a comparison of data from gill and fyke net samplings at the Swedish west coast did not reveal consistent differences in biodiversity metrics, and the statistical approaches were chosen to minimize this potential bias when comparing SR among sub-basins (Bergström et al., 2013, Chao et al., 2020; see also search of additional data sources below, and Sects. 2.3 and 4).*"
- L400: We will change accordingly.
- L417: We suggest to edit to: "*As anticipated based on earlier Baltic Sea studies on fish (e.g. Hiddink & Coleby, 2012; Ojaveer et al., 2010; Olsson et al., 2012) and other organism groups (e.g. Broman et al., 2019; Zettler et al., 2014), coastal fish SR was positively correlated with salinity (Fig. 2a, Table S), with fish SR increasing ca. threefold together with the ca. 12-fold increase in salinity (Table 2).*"
- L440: We agree and will change accordingly, i.e. remove the statement with a lag-period.
- L460: We suggest that the suggested place may not be the optimal place to add reference to the phytoplankton trend, since we here focus on prey items of benthic fish. However, we are referring to this study in the introduction.
- L463: We agree that this wording could be improved, and suggest a rewrite to: "*Taken together, these patterns suggest that the process of benthic-pelagic coupling through fish predation likely involves a lower number of species links, or functional redundancy, towards lower salinity sub-basins.*"
- L483-484: This is an interesting question, i.e. what the net effect of simultaneous warming and upper layer freshening on fish SR may be. However, given that we did not study warming, we suggest not to include further discussion on potential implications at this point.

We thank the reviewers and Editors for their efforts with this work, and are looking forward to hear from you about our manuscript.

Yours sincerely,

Birgit Koehler and co-authors

---

## Author Response (AR1)

We thank the reviewer for the thorough and constructive review of our manuscript "Species richness and functional attributes of fish assemblages across a large-scale salinity gradient in shallow coastal areas". The assessment and comments are very helpful, and we agree with the suggested changes and needed clarifications/elaborations to improve the manuscript. Please find our specific responses below.

**General comments**

Concerning the general comments, we agree it benefits the clarity of the manuscript to remove the offshore data, given that only for one sub-basin there was enough data to conduct the statistical (rarefaction-extrapolation) analysis of fish species richness. Regarding the specific comment concerning the title, i.e. that only coastal but not offshore areas were mentioned, this was solved by excluding the offshore data, following the reviewers suggestion.

We agree that the potential factors influencing fish SR should be mentioned already in the introduction. We therefore revised the following sentence in the introduction to include that aspect: "*The species composition of fish in the Baltic Sea is regulated by salinity as well (Olsson et al., 2012; Pekcan-Hekim et al., 2016), even though other factors, such as temperature or habitat complexity, might also influence large-scale patterns of fish SR in estuaries (Vasconcelos et al., 2015; Schubert et al., 2011).*"

Further, we added two new discussion paragraphs where we discuss additional factors, specifically cumulative human pressure and habitat complexity (please see P21/L436 to P22/L455).

We agree that further detail, explanation and clarification would benefit section 2.3, "Analysis of species richness data", and conducted a careful major revision of that section to better elaborate and clarify the statistical terms and methods (please see P6/l155 to P7/L188). In working on these revisions, we noted an unfortunate error in that the values of sample coverage had been incorrectly termed inventory completeness in the initially submitted manuscript (table and text). We corrected this, better explained the terms (following the reviewers suggestion), and included the values of inventory completeness (besides sample coverage) in Table 2.

We agree as well that a more detailed discussion of the fish functional characteristics and changes across the geographic gradient was warranted, and therefore expanded and revised the respective discussion section in response to the reviewers comment (please see P22/L468 to P23/L492).

**Specific comments**

We thank the reviewer for pointing out the needed technical corrections and suggestions for improving tables, and followed all of them.

Reflecting the revisions conducted to the manuscript we further slightly revised and adapted the Abstract.

We thank the reviewer and Editors for their efforts with this work, and are looking forward to hear from you about our manuscript.

Yours sincerely,

Birgit Koehler and co-authors

We thank the reviewer for the thorough and constructive review of our manuscript "Species richness and functional attributes of fish assemblages across a large-scale salinity gradient in shallow coastal areas". The assessment and comments are very helpful, and we overall agree with the suggested changes and needed clarifications/elaborations to improve the manuscript. Please find our specific responses below.

1) Manuscript structure, tables and figures

The reviewer suggests the salinity map to be shown only once in the manuscript. We agree and, to accomplish this while maintaining the necessary information, we moved Figs. 4-6 into the Supplement, and instead lifted the previous Figs. S2 and S3 into the manuscript. These show the relationships between functional groups and salinity as bi-plots, with regression lines when significant. At the same time, we completed these correlation figures by including all functional groups, where we previously (i.e. in the Supplement) had only included a subset of groups. This will facilitate complete overview over the found relationships when most of the previous maps were moved to the Supplement. By these changes, the number of figures in the manuscript was reduced from previously six to five. Further, the acronyms of the sub-basins, which are given in the bar-plots of Fig. 3, are now defined in the figure legend. We also followed the reviewers suggestion to place a larger-scale map for reference into the maps showing the Baltic Sea.

Moreover, we moved Table 3, which gives the observed, standardized and estimated Shannon and Simpson Diversity, as well as Table 4, which gives regression information for relationships between Shannon and Simpson Diversity with salinity and temperature, from the manuscript into the Supplement. These revisions accommodate the comment that the three estimates (std, obs, est) are, for the more common species, very similar. Fig. 2 still shows both the values (referring to Table S5) and relationships with salinity (referring to Table S6). By these revisions, the number of Tables in the manuscript was reduced from previously five to three. Also, the tables as such have become considerably simpler following the exclusion of the offshore data, as suggested by the reviewer.

In Table 2, i.e. for species richness, we would like to keep all three estimates (i.e. std, obs and est), as these differ more than for ShD and SiD since the rare species are included. One of our central messages, i.e. that fish SR decreased about three-fold from highest to the lowest salinity, is based on $SR_{std}$. It is also relevant to compare $SR_{est}$ with e.g. total observed SR (i.e. based on incidence data plus presence observations). For these reasons we would also like to keep Table 2 in form of a Table, such that the precise values (and error estimates) can be directly compared. To accommodate the reviewers comment, we better explained the three different estimates (obs, std and est) in section 2.3, "Analysis of species richness data". In short, the purpose of the standardised estimate is to enable accurate comparisons among areas in spite of differing sample sizes and inventory completeness. The purpose of the estimated SR is to extrapolate to a likely number of species in an area, if inventory was continued. The complementarity of the different estimates is discussed in the last sentence of the first discussion paragraph, and in the second paragraph of the discussion.

2) Coastal resident vs. migrating species

Since the marine migrating and visiting species are part of the coastal fish assemblage during parts of the year, and depend on coastal areas during parts of their life cycle, we included them in the statistical analyses. The mix of resident and mobile fish species indicates the dynamic nature of certain coastal areas, and their connectivity with the open sea. We agree with the reviewer that inclusion or exclusion of marine migrating and visiting fish species in the analyses does affect how the inventory completeness (IC) turns out, but find that including them results in a more complete representation of SR. In response to this comment, we added

a note in the respective Methods section (Sect. 2.3) which details that both resident and migrating/visiting species were included during calculation of IC (P7/L175), and added discussion on this aspect (P20/L374 to P20/L376).

We further agree with the reviewer that a lower or higher proportion of marine migrating and visiting species would influence the estimate of IC, and that changes in this proportion over time would affect the IC estimate. Adding such a temporal perspective would be interesting to follow up in a future study, but was not included in this large-scale spatial study where we merged monitoring data from many years to obtain as accurate as possible spatial comparisons within the limits of available data. It is important to note, though, that the sample coverage (SC) in our study, i.e. the inventory completeness of the more common species, is not strongly affected by the incidence frequency of rare species, and was very similar across sub-basins (Table 2). Since $SR_{std}$ and $SR_{est}$ depend on SC they are rather robust against changes in the proportion of (more rarely present) migrating/visiting vs. resident species, and hence against including or excluding migrating and visiting species. In response to this reviewers comment, we added discussion on this aspect (P22/L445 to P22/L449).

3) Offshore data

We agree that it benefits the general clarity of manuscript to exclude the offshore data, particularly given that only one sub-basin had enough data to conduct the statistical (rarefaction-extrapolation) analysis of fish SR. Further, as the reviewer assumed, data from the BITS survey carried out in the Baltic Sea were indeed not included due to a lack of data for shallower depths (i.e. <30 m).

4) Discussion on other drivers

We agree with the reviewer, and included two new discussion paragraphs on the potential influence of other drivers, particularly on cumulative human pressure and habitat complexity (please see P21/L436 to P22/L455).

5) Discussion on functional attributes (traits)

We agree that the fish functional characteristics and their changes along the salinity gradient warranted more detailed discussion, that the statement on benthic-pelagic coupling needed to be clarified, and that it will be interesting to couple the discussion to ecosystem processes. To meet these comments, we revised and expanded the respective discussion paragraphs (please see P22/L468 to P23/L492).

Concerning the other comments:
- L32: We refer to species richness, and clarified the statement accordingly (P2/L30-33).
- L49: We added "on average" to accommodate the fact that certain regions experience less intense water cycling, while the general trend is an intensified hydrological cycle.
- L78: This comment does not anymore apply since offshore areas were, upon suggestion from the reviewer, removed from the manuscript.
- L116: In separated ecosystems, such as islands or lakes, SR usually increases with area ("species-area relationship"). However, such separation is not the case for our coastal sub-basins here. While the studied sub-basins are hydrographically distinct, with water exchange being separated to a certain degree by shallow sounds or sills, they are still connected. Based on this reviewers comment we realised that giving the size of the shallow coastal areas in table and Methods may cause confusion in this regard. We therefore removed these values, which are also not used further in the

study, leading to simplification of tables (as suggested would be useful by the reviewer).

- L128: We revised the respective text part (P3/L130-134).
- L139: We revised this part to better explain the motivation for the chosen cutoff (P5/L135 to P6/L142). The cutoff was essentially based on that we found that sub-basins with less than one hundred fish species incidences had too little data for statistical analysis. During revision in response to this comment we realized an additional aspect that needed to be clarified in the manuscript. Specifically, while we extracted all available data from the database covering nearly five decades as stated in the manuscript (1975-2021), only one sub-basin had annual samplings during all those years, while most sub-basins had data from 17+ years, and a few sub-basins had data from less years (i.e. 4 and 9 years, excluded from statistical analysis based on that, see above). We edited the text accordingly, and added a new Supplementary Table detailing for which years fishing data was available per sub-basin (new Table S2).
- L110-147: The incidence data is official, quality-controlled survey data for which species are correctly identified by taxonomic specialists. As an additional double-check of uncommon species we used the HELCOM list of macro-species in the Baltic Sea. Concerning the "observation databases", where e.g. citizens can report species observations and which hence are less reliable, we did a careful cross-check where unreasonable occurrences were considered falsely identified and discarded. We elaborated the respective text pieces to clarify these aspects, please see section 2.2.
- L167: The comment made us realise that this text part could be confusing since the rarefaction-extrapolation method is called "Chao Richness method", and the used R-function has the same name but includes several different calculations. We conducted a major revision of Sect. 2.3 to accommodate this and other comments on it.
- L170: We suggest to clarify that the observed values were standardised to the minimum SC. The purpose of $SR_{std}$ is to allow for accurate comparisons between sub-basins, given that all standardised values give SR for the same SC, hence representing an estimate which is not biased by how completely the compared areas were sampled. During revision, we noted an unfortunate mistake in that "sample coverage" (SC) was erroneously called "inventory completeness" (IC). This is now corrected, and better explained in the revised Sect. 2.3. We agree with the reviewer that SC was high and very similar across sub-basins, varying between 98.5-99.9%, and hence in the case of this study did not strongly influence the obtained values. The correction is still needed for accurate, unbiased comparisons between sub-basins, and we therefore find it important to keep the standardisation in the manuscript rather than moving it to the Supplement. However, we understand the need of presenting the data and results more clearly, especially in the Tables. For this aim, we excluded the offshore data, which considerably simplified text and tables, and reduced the number of tables and figures in the manuscript (as suggested by the reviewer, and please see above for details). To further accommodate this reviewers comment and the comment above we also conducted a major revision of Sect. 2.3.
- L222: This comment does not apply anymore following omission of the offshore data, as suggested by the reviewer.
- L237: This part was changed, because we realized during revision that we had by mistake interchanged the terms inventory completeness and sample coverage. Now, inventory completeness shows a larger variation, and we would like to refer to Table 2 rather than translating the percentages into species numbers.

- L250: Extrapolation is recommended up to maximally twice the actual sample size for SR (Chao et al., 2020). We now included this information in the respective methods section (Sect. 2.3), and in the legend of Fig. 1.
- Table 3. We moved this table to the Supplement (please also see above). A graphic illustration of the trends across the salinity gradient is available in Fig. 2, and, for ShD and SiD, the observed values are very similar to the standardized and estimated values, given that the missing species represent rare species. Hence, for study sites with high SC, the observed ShD and SiD are already largely unbiased estimates. To further accommodate the reviewers general comment that the manuscript contained too many tables and figures we also moved Table 4 to the Supplement (please see above).
- Table 5. We now included the number of observations in Table 3, and in the legend of Table S6.
- Previous Figs. 4-6. We agree it was redundant to repeat the salinity map repeatedly in these figures. We moved previous Figs. 4-6 to the Supplement, and instead lifted previous Figs. S2 and S3 to the manuscript. These figures directly show the relationships between functional groups (on habitat use and vertical distribution) and salinity.
- L372: Mora et al., 2008 applied rarefaction-extrapolation methods similar to the one used in our study, but used different statistical models. Appeltans et al. 2012 based their IC estimates on a statistical model based on historical rates of species description. While the methods differ more or less from the one we used the estimated property is the same, i.e. IC as the proportion of observed to total species, which motivates our comparison in spite of methodological variation.
- L379: We rewrote this sentence to clarify which ratio we mean, moved it to the Results (end of Sect. 3.5), but also took it up as discussion point in this place. We rewrote to "calculated based on data presented in Table 2" to make clear that the ratio is not given in the Table, but that the values needed to calculate it are found there.
- L394: We understand that this statement could be confusing, and rewrote it accordingly (P20/L393 to P20/L396).
- L400-414: We now added more information on this aspect already in Sect. 2.2 (P5/L130-134).
- L400: We changed accordingly.
- L417: We edited the sentence accordingly (P21/L414-416).
- L440: We agree and removed the statement with a lag-period.
- L460: We suggest that this may not be the optimal place to add reference to the phytoplankton trend, since we here focus on prey items of benthic fish. However, we are referring to this study in the introduction.
- L463: We agree that this wording could be improved, and rewrote it (P23/L486-487).
- L483-484: This is an interesting question, i.e. what the net effect of simultaneous warming and upper layer freshening on fish SR may be. Given that we did not study warming, however, we did not include further discussion on potential implications at this point.

We thank the reviewers and Editors for their efforts with this work, and are looking forward to hear from you about our manuscript.

Yours sincerely,

Birgit Koehler and co-authors

---

## Author Response (AR2)

Dear Dr. Marañón,

thank you for serving as Editor for our manuscript "Species richness and functional attributes of fish assemblages across a large-scale salinity gradient in shallow coastal areas".

Please find below our details on how the manuscript was revised to address the reviewer's comments (which are kept in italic style).

*This is the second round of review for Koelher et al. I thank the authors for addressing and answering in detail the reviewers' comments. I think that the authors have done good work reviewing the manuscript and that this has helped make the manuscript more robust and streamlined. I still have a few minor comments for the authors to consider. Please note that I refer to the Lines from the manuscript with track changes, sorry for the confusion.*

*Main:*

*- Figure 2- and table 3: why the expectation of linear regression? how are the residuals looking? (any patterns?). In many cases it looks like a non-linear relationship would be more appropriate, maybe some GAMs with restricted k smoothing factors. For example, Fig 4c. SR obs does not seem to follow a linear relationship (residuals not homogeneously distributed, all observed SR at low salinity are under the regression line)*

Here, we wanted to conduct univariate statistical testing on the different SR estimates (overall and for the functional groups, as well as observed, standardized and estimated SR) vs. salinity (and sometimes temperature). To address the aspect of potential non-linearity, the distributions of the variables were carefully assessed, and transformations (log10 or Yeo-Johnson, see Sect. 2.6 "Statistical analyses") applied to best meet the requirement of normally distributed residuals. The resulting residual distributions were assessed using standard residual diagnostic plots in R, assuring that no pronounced residual patterns remained.

Concerning Fig. 4c, only part of the green data points is well visible, as they are very similar to the other (red and blue) data points and overlying each other in some cases. We added a note in all respective figure legends stating that data points, lines and confidence intervals are in some cases overlying each other, and referring to the Table with regression results and to the Supplementary Table with raw data for details. The residuals of Fig. 4c did not violate the assumption of homogeneously distributed residuals.

*- Incidence (e.g. L83 and throughout). The term "incidence" is not defined. Is an incidence an observation / occurrence? If so, then why put in opposition with "presence" data (e.g. L460). When reading I thought that the term was sometimes confusing, notably when calculating % incidence of undetected species. You could define what is "incidence information" the first time the word is used.*

We understand that we did not in all cases distinguish data sources/formats and terms clearly enough. We separate between: data from systematic samplings, where all caught species were determined to species and the precise sampling locations were known (these were entered into the "fish incidence database"; Sect 2.2), and: data from additional sources, where only

presence/occurrence was reported. As these additional data do not stem from systematic sampling efforts they could not be used in the statistical analyses (see Sect 2.3: "These "additional data sources" were used as complementary information on $SR_{obs}$ but could not be used in the statistical analysis since they did not have complete sampling and species incidence information.").

To accommodate this comment on remaining ambiguity in terminology in some places, we are now more specific on the term "incidence". We included an explanation/definition in the concerned Methods Section, Sect. 2.3. Further, in cases where we previously used the term "presence" in relation to the "additional data sources", we now use "species records" instead. Last, we have made sure that we use the same term (i.e. "incidence data plus additional data sources", as in Table 2) when referring to this data.

*Others:*

*- L81: definition of coastal areas? distance from the shore? For IBTS data, which criteria is used to be defined as coastal sampling?*

Coastal areas were defined in alignment with the EU Marine Strategy Framework directive (also corresponding with biological parameters for the EU Water Framework Directive), and obtained from the Water Information System of Sweden. We added the definition and source into the text. We also clarified that the same geographical criteria was used for IBTS data.

*- L32-33: the link between the first and second part of the sentence does not follow a logical flow, consider separating and rephrasing.*

We separated the sentences, and edited this text part.

*- L130 "Geographical delineation". I might have missed it, but which geographical delineation are you referring to?*

This refers to the statement above (L117/118): "Coastal areas were delineated using official national

definition". We added "of shallow coastal areas" in the pointed out place for clarity.

*- L165: This is done per sub-basin?*

Correct. We have edited accordingly.

*- Table 2 "species incidence" and footnote a. Is this the sum of all samples' unique species richness? (Referring back to comment earlier on incidence).*

This is the total number of unique species across sampling occasions. We have edited the footnote for clarity.

*Footnote c also needs clarification.*

We think this is clarified both by defining the term "incidence" in response to the reviewers second main comment, and by editing the footnote "b", since footnote c is similar but for observed and estimated incidences.

*- Table 3: There are so many relationships tested that the p-value has little meaning, you could consider adjusting p-values for multiple comparisons, or delete the p-value, the R2 on its own can give support on whether there is a clear pattern or not. I am not a big fan of p-value, but would of course understand if the authors want to keep it.*

These are the results from simple linear regressions, i.e. between salinity and fish SR in the different functional groups, one group at a time. Therefore, multiple comparison adjustments of the test statistics (and P-values) were not required (no multiple testing conducted). We edited the table legend for clarity in this respect, and like to keep the P-values (in addition to the $R^2$ values that are also included in the table).

*- L275: instead of "exist" maybe "present in the sub-basins"*

We edited as suggested.

*- Fig 1. could you increase the height of the figure? small height gives a wrong impression that all curves have reached the asymptote.*

We followed the reviewers suggestion and edited the height-width ratio of the figure for improved visualisation.

*- L438: So more concretely, the IC suggests that how many species are probably occurring but not observed, 1-2 species still not identified? More?*

The IC gives the proportion of species detected from the estimated "true" SR (explained in Sect. 2.3). Hence, 100% - IC gives the percentage of species statistically likely to exist but not detected in the sampling. We understand that the reviewer would like to "translate" the IC to "number of undetected species", i.e. $SR_{est}$ minus $SR_{obs}$. We edited in this information.

*- L445: this sentence could be improved: "which are only more rarely present"?*

We agree, and edited the respective sentence.

*- L451: Based on your results I would say that SR was also well described for rare species? And that few, extremely rare species might still not be listed*

We generally agree. This is also stated in Sect 3.3.: "The SC exceeded 98.5% in all sub-basins (Table 2), suggesting that these undetected species were highly rare, likely representing <1.5% of incidences." The argument here is based on that $SR_{obs}$ was similar to $SR_{est}$ for Shannon diversity (number of frequent species) and Simpson diversity (number of highly frequent species; Table S5). In contrast, the total $SR_{obs}$ was lower than the total $SR_{est}$ by 1-37 species, depending on sub-basin, i.e. this number of (highly) rare species likely remained undetected. Hence, the SR of all species (including rare species) was not as well described as the SR of frequent and highly frequent species.

To accommodate this comment and clarify the argument we moved this respective paragraph higher up in the discussion. We included the information how many rare species (in numbers) were statistically estimated as undetected based on the incidence data. This discussion aspect is then immediately followed by the statement that the species lists were essentially complete when additional data sources were considered, and explicitly edited to name the rare species here again. We think that this revision gives a better flow and takes care of the raised aspect. Further, we edited in Sect. 3.5 from "rare and very rare" to "highly rare", and similarly in the concerned paragraph of the discussion.

*- L462: "50-90%" where does this range comes from? Maybe should be written in the results section.*

We now clarify in brackets how this ratio is calculated. The values are presented in the results, but we like to highlight this comparison in the discussion.

*"are found" (= are occurring in the area) or 50-90% were "observed in the samples"? If the last one, then shouldn't it be "at least 50-90% …"?*

Here, as now specified in response to the comment (see above), we compare $SR_{est}$ with the HELCOM species numbers per sub-basin. Our $SR_{est}$ is corrected for sample size. The $SR_{obs}$ from HELCOM is not, and may hence be a lower bound estimate. Therefore, the here estimated ratio might be biased somewhat high, which is why we wrote "ca.". We wrote "are found" since it concerns the estimated SR, i.e. the sample-size corrected species richness.

*- L516: "expected" instead of indicated? you didn't test*

We agree and changed the wording as suggested.

*- SC: SC, throughout the text for simplicity maybe only write "sample coverage" without the abbreviation as it is not consistently used*

We think the acronym is useful, since it is commonly used, and also used repeatedly throughout the manuscript. In some places the word was spelled out for specific reasons, e.g. in the Abstract prior to definition, or in a Figure legend, but otherwise the acronym is used consistently. We like to keep it.

*- L545: do you rather mean "functionally-driven" differences?*

We removed this word from the sentence, as it was clear without.